


# The Iso2k Database: A global compilation of paleo-δ¹⁸O and δ²H records to aid understanding of Common Era climate

Bronwen L. Konecky[1], Nicholas P. McKay[2], Olga V. Churakova (Sidorova)[3], Laia Comas-Bru[4], Emilie P. Dassié[5], Kristine L. DeLong[6], Georgina M. Falster[1], Matt J. Fischer[7], Matthew D. Jones[8], Lukas Jonkers[9], Darrell S. Kaufman[2], Guillaume Leduc[10], Shreyas R. Managave[11], Belen Martrat[12], Thomas Opel[13], Anais J. Orsi[14], Judson W. Partin[15], Hussein R. Sayani[16], Elizabeth K. Thomas[17], Diane M. Thompson[18], Jonathan J. Tyler[19], Nerilie J. Abram[20], Alyssa R. Atwood[21], Jessica L. Conroy[22], Zoltán Kern[23], Trevor J. Porter[24], Samantha L. Stevenson[25], Lucien von Gunten[26], and the Iso2k Project Members*

[1]Department of Earth and Planetary Sciences, Washington University, Saint Louis, Missouri, 63108, USA
[2]School of Earth and Sustainability, Northern Arizona University, Flagstaff, AZ, 86011, USA
[3]Institute of Ecology and Geography, Siberian Federal University, Krasnoyarsk, 660041, Russian Federation & Department of Forest Dynamics, Swiss Federal Institute for Forest, Snow and Landscape Research WSL, Birmensdorf, CH-8903, Switzerland
[4]School of Archaeology, Geography & Environmental Sciences, University of Reading, Reading, Berkshire, United Kingdom
[5]EPOC Laboratory, University of Bordeaux, France, 33615, France
[6]Department of Geography and Anthropology, Coastal Studies Institute, Louisiana State University, Baton Rouge, LA, 70803, USA
[7]NSTLI Environment, ANSTO, Sydney, NSW, 2234, Australia
[8]School of Geography, University of Nottingham, Nottingham, NG7 2RD, UK
[9]MARUM Center for Marine Environmental Sciences, Bremen University, Bremen, 28359, Germany
[10]Aix Marseille University, CNRS, IRD, INRAE, Coll France, CEREGE, Aix-en-Provence, 13545, France
[11]Earth and Climate Science, Indian Institute of Science Education and Research, Pune, Maharashtra, 411008, India
[12]Department of Environmental Chemistry, Spanish Council for Scientific Research (CSIC), Institute of Environmental Assessment and Water Research (IDAEA), Barcelona, Barcelona, 08034, Spain
[13]Polar Terrestrial Environmental Systems and PALICE Helmholtz Young Investigator Group, Alfred Wegener Institute Helmholtz Centre for Polar and Marine Research, Potsdam, 14473, Germany
[14]L-IPSL, CEA-CNRS-UVSQ-Université Paris Saclay, Laboratoire des Sciences du Climat et de L'Environnement, Gif Sur Yvette, 91191, France
[15]Institute for Geophysics, University of Texas at Austin, Austin, TX, 78758, USA
[16]School of Earth and Atmospheric Science, Georgia Institute of Technology, Atlanta, GA, 30332, USA
[17]Department of Geology, University at Buffalo, Buffalo, NY, 14260, USA
[18]Department of Geosciences, University of Arizona, Tucson, Arizona, 85719, USA
[19]Department of Earth Sciences, The University of Adelaide, Adelaide, South Australia, 5005, Australia
[20]Research School of Earth Sciences and Centre of Excellence for Climate Extremes, Australian National University, Canberra, ACT, 2601, Australia
[21]Department of Earth, Ocean, and Atmospheric Sciences, Florida State University, Tallahassee, Florida, 32306, USA
[22]Department of Geology, University of Illinois at Urbana-Champaign, Urbana, IL, 61822, USA
[23]Institute for Geological and Geochemical Research, Research Centre for Astronomy and Earth Sciences, MTA Centre for Excellence, Budapest, H-1112, Hungary



[24]Department of Geography, University of Toronto - Mississauga, Mississauga, Ontario, L5L1C6, Canada
[25]Bren School of Environmental Science & Management, University of California, Santa Barbara, Santa Barbara, CA, 93106, USA
[26]PAGES International Project Office, Bern, 3012, Switzerland

*Correspondence to*: Bronwen L. Konecky (bkonecky@wustl.edu)

*A full list of authors appears at the end of the paper.

**Abstract.** Reconstructions of global hydroclimate during the Common Era (CE; the past ~2,000 years) are important for
providing context for current and future global environmental change. Stable isotope ratios in water are quantitative
indicators of hydroclimate on regional to global scales, and these signals are encoded in a wide range of natural geologic
archives. Here we present the Iso2k database, a global compilation of previously published datasets from a variety of natural
archives that record the stable oxygen ($\delta^{18}O$) or hydrogen ($\delta^2H$) isotopic composition of environmental waters, which reflect
hydroclimate changes over the CE. The Iso2k database contains 756 isotope records from the terrestrial and marine realms,
including: glacier and ground ice (205); speleothems (68); corals, sclerosponges, and mollusks (145); wood (81); lake
sediments and other terrestrial sediments (e.g., loess) (158); and marine sediments (99). Individual datasets have temporal
resolutions ranging from sub-annual to centennial, and include chronological data where available. A fundamental feature of
the database is its comprehensive metadata, which will assist both experts and non-experts in the interpretation of each
record and in data synthesis. Key metadata fields have standardized vocabularies to facilitate comparisons across diverse
archives and with climate model simulated fields. This is the first global-scale collection of water isotope proxy records from
multiple types of geological and biological archives. It is suitable for evaluating hydroclimate processes through time and
space using large-scale synthesis, model-data intercomparison and (paleo)data assimilation. The Iso2k database is available
for download at: https://doi.org/10.6084/m9.figshare.11553162 (McKay and Konecky, 2020).

**1. Introduction**

**1.1 Progress and challenges in the synthesis of Common Era hydroclimate**

The past ~2,000 years, otherwise known as the Common Era (CE), are an important research target for contextualizing
modern climate change. Decades of paleoclimate research have yielded numerous records spanning all or part of this time
period, making it sufficiently data-rich to assess the range of natural (internal and forced) climate variability prior to the
industrial revolution. These records are also used in conjunction with climate model simulations to detect and attribute
anthropogenic climate change. Over the past several years, large-scale data synthesis efforts within the international
paleoclimate community have produced important constraints on regional to global surface air and ocean temperature
patterns during the CE (McGregor et al., 2015; McKay and Kaufman, 2014; PAGES 2k Consortium, 2013, 2017, 2019;
Tierney et al., 2015). However, progress on the synthesis of hydroclimate patterns has been limited (PAGES Hydro2k



Consortium, 2017), despite the societal relevance of the changing water cycle (e.g., Kelley et al., 2015). The water cycle is a far more complex target than surface air and ocean temperature, and different proxy systems track different aspects of the water cycle in different ways (PAGES Hydro2k Consortium, 2017). For example, annual precipitation amount at any given location on the Earth's surface is governed not just by atmospheric processes that deliver moisture to the region, but also by topography, varying characteristics of storms and associated clouds, dynamics of the seasonal cycle, and variations in the

contribution of extreme precipitation events (Bowen et al., 2019).

Individual paleoclimatic proxy types are often sensitive to multiple aspects of the water cycle that can be difficult to disentangle, making it challenging to directly compare among proxy types. For example, precipitation amount in the Arctic could be inferred from two common precipitation proxies: grain size from lake sediments and accumulation rates from ice

cores. Grain size fluctuations in lake sediments can track extreme precipitation and runoff events, but inter-lake comparison requires knowledge of lake morphometry and competing moisture source regions (Conroy et al., 2008; Kiefer and Karamperidou, 2019; Rodysill et al., 2019). Comparison of sedimentary grain size to snow accumulation rates would be uninformative without understanding how annual precipitation and dry season ablation, which both affect accumulation rates, are related to moisture delivery from extreme precipitation events (Hurley et al., 2016; Thompson et al., 1986). Snow

accumulation rates can be strongly affected by air temperature, whereas grain size is generally not. Thus, although comparison of such heterogeneous hydroclimatic proxies is certainly possible, the lack of a common environmental signal to serve as a reconstruction target has been a major hindrance to the global reconstruction of hydroclimatic variables. These challenges have been further exacerbated by archive- and record-specific standards for data formatting, sampling resolution, metadata availability, and public archiving. These limitations may be addressed by creating a metadata-rich, multi-proxy,

and multi-archive database of hydrological proxies united through standardized formatting and a common environmental signal: water isotopes.

**1.2 The potential for a network of paleo-water isotope records to track past hydroclimate variations**

In order to address these challenges, we focus here on the stable oxygen ($\delta^{18}$O) and hydrogen ($\delta^2$H) isotopic compositions of

environmental waters such as precipitation, seawater, lake water, and soil and groundwater **[Figure 1]**. The stable isotopic compositions of such waters (here collectively referred to as "water isotopes") have long been used as integrative tracers of the modern water cycle (e.g., Bowen et al., 2019; Galewsky et al., 2016; Gat, 2010; Rozanski et al., 1993). The rare heavy isotopologues of water (e.g., $^1$H$_2$$^{18}$O, $^1$H$^2$H$^{16}$O) fractionate from their lighter, more common counterpart ($^1$H$_2$$^{16}$O) during evaporation, condensation, and other phase changes, capturing an integrative history of parcels of water as they move

through and among oceans, atmosphere, and land **[Figure 1]**. Global databases of isotopic measurements of modern precipitation (IAEA/WMO, 2019), rivers (Halder et al., 2015), seawater (LeGrande and Schmidt, 2006), and water vapor (Galewsky et al., 2016) have contributed considerably to our understanding of the contemporary water cycle on scales from micro (e.g., cloud microphysics) (Kurita et al., 2011) to mesoscale (e.g., hurricane dynamics) (Good et al., 2014; Kurita et





al., 2011) to global (e.g., residence time of atmospheric moisture) (Aggarwal et al., 2012). More recently, space-borne

measurements of $^1$H$^2$HO/$^1$H$_2$O in multiple levels in the atmosphere have identified the critical role of poorly-observed

processes such as tropical rain re-evaporation (Aggarwal et al., 2012; Worden et al., 2007) and forest-atmospheric feedbacks

(Wright et al., 2017). Together with climate and Earth system model simulations, which increasingly incorporate

sophisticated water isotope tracers into their hydrologic schemes (Brady et al., 2019; Haese et al., 2013), water isotopes offer

observational constraints on processes that are otherwise difficult to identify or constrain (Brady et al., 2019; Nusbaumer et

al., 2017).

In the paleoclimate realm, hydroclimate proxy records using water isotopes are commonly obtained from a variety of natural

archives, including glaciers, ground ice, cave formations, corals, sclerosponges, mollusk shells, tree wood, lake sediments,

and marine sediments. Of all of the proxy types that are used to reconstruct past hydroclimate changes, water isotopes are

arguably the most common, and certainly the most widely distributed geographically. A global, spatially distributed network

of water isotope proxy records therefore has the potential to capture features of large-scale circulation patterns while

minimizing site-specific influences from individual locations (Konecky et al., 2019b). Paired with an understanding of water

cycle processes from modern observations and isotope-enabled model simulations, reconstructions of paleo-$\delta^{18}$O and $\delta^2$H

from these archives can provide critical information about moisture source and air mass transport history, precipitation

characteristics, glacial ice volume changes, and temperature prior to the beginning of instrumental climate observations.

Further, Proxy System Models (Evans et al., 2013) are available for most water isotope proxies, facilitating direct

comparison with paleoclimate model output and thus an improved understanding of the climate dynamics responsible for

observed (spatial and temporal) water isotope variability (Dee et al., 2015, 2018; Jones and Dee, 2018; Konecky et al.,

2019a; Thompson et al., 2011).


One of the obstacles to synthesizing hydroclimate-sensitive paleoclimate records has been a lack of standardized metadata at

the proxy system level that systematically encodes important variables necessary for both integrating records into a multi-

proxy synthesis, and interpreting the results. Although the paleoclimate community is in the process of defining and adopting

metadata conventions (Khider et al., 2019), the 'bare minimum' current standards (e.g., ISO 19115 for geographic metadata)

used by World Data System (WDS) repositories (e.g., NOAA Paleoclimatology, PANGAEA) are insufficient for

characterizing water isotope proxy systems in a way that can be reliably applied to large-scale paleo-hydroclimate syntheses.

One key example of this challenge is the temperature dependence of O- and H-isotopic fractionation, which has frequently

been exploited to reconstruct past temperature changes in locations where air or water temperature exerts first-order

influence on isotope ratios in precipitation and/or seawater (Kilbourne et al., 2008; Meyer et al., 2015; Porter et al., 2014).

Yet in most places, the influence of temperature on isotopic fractionation is only one of many factors that influence the $\delta^{18}$O

and $\delta^2$H of precipitation (Liu et al., 2012; Thomas et al., 2018) and seawater (Cobb et al., 2003; Partin et al., 2012). A

network of water isotope records will inevitably contain information about air and water temperature, but also other key



hydroclimatic variables such as atmospheric moisture source changes and surface water evaporation. In order to tap the full potential of water isotope proxy records in a large-scale synthesis, metadata associated with such records must be sufficient

to capture at least a bare minimum of the complexity of the environmental signals that the records contain.

Additional metadata challenges have hindered progress in paleo-water isotope synthesis thus far. Most published datasets shared outside WDS repositories follow non-uniform metadata standards or contain minimal metadata. Datasets are often catalogued using different conventions (often at the authors' discretion), stored in varying formats (e.g., text, CSV, PDF),

and uploaded to different public or private (i.e., behind journal paywalls) repositories. Furthermore, datasets are frequently archived without the raw chronological information that would be required to propagate age uncertainties if desired. These challenges are common to any paleoclimate synthesis effort and are not unique to water isotopes (Atsawawaranunt et al., 2018; Emile-Geay and Eshleman, 2013; PAGES 2k Consortium, 2017), but they exacerbate the challenge of hydroclimate-specific metadata needs.


### 1.3 The PAGES Iso2k database

Here we introduce the Past Global Changes (PAGES) Iso2k database, a collection of 756 water isotope proxy records (i.e., individual time series) from 505 sites (geographic locations) covering all or part of the CE. The database has been assembled by the PAGES Iso2k Project (hereafter "Iso2k"). The Iso2k database contains $\delta^{18}O$ and $\delta^{2}H$-based paleoclimate records from

ten different archives: glacier and ground ice (205 records); speleothems (68 records); corals, sclerosponges, and mollusks (145 records); wood (81 records); terrestrial and lake sediments (158 records); and marine sediments (99 records). Of these, 606 records are considered to be primary time series for each site **[Figure 2]** (see Section 2.4 and Supplementary Table 1). To address the complexity of environmental signals preserved in these proxy records, the database contains detailed metadata about each record's isotope systematics and proxy system context, as well as details about the original authors'

climatic interpretation, chronological and analytical uncertainties, and other information required for robust data synthesis and interpretation. Iso2k has developed a uniform framework suitable for all proxy archives in the database. The architecture of the Iso2k database therefore provides a scalable foundation on which future multi-proxy hydroclimatic databases can be built, for example incorporating non-isotopic proxy records such as the grain size and ice accumulation example in section 1.1.


The Iso2k database is the latest in a series of community-led paleoclimate data synthesis efforts endorsed by PAGES (Atsawawaranunt et al., 2018; Kaufman et al., in revision; McGregor et al., 2015; McKay and Kaufman, 2014; PAGES 2k Consortium, 2013, 2017; Tierney et al., 2015). The main distinguishing feature of the Iso2k database is that it is not organized around one archive type, climate variable, or region; rather, it contains a systematic representation of the suite of

environmental signals preserved in the water isotopic composition of diverse paleoclimatic archives, with no *a priori* assumptions about the underlying climatic interpretation of those signals. This novel approach yields a database that is



flexible enough to evaluate many different environmental parameters and processes during the CE, depending on investigator interest. The Iso2k database also contains even more comprehensive metadata descriptions compared with previous PAGES compilations (e.g., PAGES 2k Consortium, 2017). Database users can therefore filter for and process only 180 the records required for their research question of interest.

This data descriptor presents version 1.0.0 of the PAGES Iso2k database. We describe the collaborative process of assembling the database (including quality control and validation), and outline the structure and contents of the database (including data selection criteria, metadata, and chronological information). All data are provided in the Linked Paleo Data 185 (LiPD) format (McKay and Emile-Geay, 2016) and are machine readable across different platforms and operating systems. We provide files with sample code to quickly explore the database using various programming languages and platforms (R, Matlab, Python). The database itself is available from https://doi.org/10.6084/m9.figshare.11553162 (McKay and Konecky, 2020) (note this beta version will become version 1.0.0 upon acceptance of this manuscript). Upon acceptance of this manuscript the database will be made available through the NOAA NCEI World Data Service for Paleoclimatology (WDS- 190 NOAA), with a landing page and links to download the serializations for R, MATLAB, and Python.

## 2. Methods

### 2.1 Collaborative model

Iso2k is a contribution to Phase 3 of the PAGES2k Network (PAGES 2k Network Coordinators, 2017). Calls for 195 participation in Iso2k were widely distributed, ensuring a representative cross-section of scientists from various disciplines (Konecky et al., 2017, 2018, 2015; Partin et al., 2015). Iso2k built on the successes and challenges of previous PAGES2k projects (Anchukaitis and McKay, 2014; Kaufman, 2014; PAGES 2k Consortium, 2017; PAGES Hydro2k Consortium, 2017) when deciding on the selection criteria (i.e., requirements for inclusion of records) and metadata fields necessary to make the database suitable for a wide range of applications. Most work was done remotely via teleconferences, with one in- 200 person meeting at the 2017 PAGES Open Science Meeting in Zaragoza, Spain.

The workload for assembling the data and metadata was subdivided among working groups, representing one of the following archive types: marine sediment, marine carbonates (corals, mollusks, sclerosponges), glacier ice, ground ice, lake sediments, speleothems, and wood. This archive-based approach ensured that data were collated by researchers with an in- 205 depth, process-based understanding of each proxy system.

### 2.2 Data aggregation and formatting

The database comprises publicly-available water isotope proxy records that span all or part of the CE and meet the criteria outlined in Section 2.3. The database was compiled in two main stages. During the first stage, the archive teams obtained





records, entered data, and compiled the extensive metadata outlined in Section 4. During the second stage, the data and metadata were extensively quality controlled following the procedure outlined in Section 2.4.

We used a variety of sources to identify records for inclusion in the database. We first extracted records that met our selection criteria (described in section 2.3.1) from existing data compilations, including the PAGES2k temperature database

(PAGES 2k Consortium, 2017), the Arctic Holocene Transitions database (PAGES 2k Consortium, 2017; Sundqvist et al., 2014), and the SISAL database (Atsawawaranunt et al., 2018). Archive teams then searched the literature and online data repositories (WDS-NOAA and PANGAEA) for additional suitable datasets. For records that had been published but not previously been made available in an online public repository (referred to as 'dark data'), datasets were digitized from publication tables, appendices, and supplementary materials. Datasets that were not available in their original publications

were requested from the authors by email. If two or more email requests went unanswered the dataset was deemed not publicly available and therefore did not meet that criterion for inclusion in this database. Most archive teams added 1–5 dark datasets to the Iso2k database. However the majority of dark data added to the Iso2k database were from the lake sediments archive. Data from more than 50 of the 158 lake and other terrestrial sediment records in the Iso2k database are available in an online public repository for the first time here.


In addition to isotopic datasets, raw age control data (e.g., $^{14}$C ages) were obtained for records where age-depth modeling is required (i.e., non annually-resolved records). Many isotopic datasets that were available through data repositories did not contain raw age control data, in which case we followed the dark data procedure described previously to obtain appropriate chronological data from the authors. For dark age control data, authors were emailed with a request for the data and a

spreadsheet template where chronological information could be added. Age control data from authors who did not respond to these requests could not be added to the database. Again, the majority of 'dark' age control data added to the Iso2k database was from the Lake Sediments archive (over 40 age control datasets).

Metadata (Section 4) were obtained from the data source, extracted from the original publication, or requested from the

original data generators. We note that even for datasets that were previously publicly available, the Iso2k database has expanded on these data by adding chronological data and compiling an extended suite of metadata not previously available in a consolidated format.

## 2.3 Record selection criteria

Criteria for inclusion in the database were formulated to optimize spatio-temporal coverage of the data, with the goal of building a comprehensive database of water isotope records that can be sub-sampled as needed to address diverse scientific questions. The selection criteria for data records to be included in the Iso2k database are as follows.





### 2.3.1 Record resolution and duration

The duration and temporal resolution of records included in the Iso2k database varies by archive type. For ~annually- or ~sub-annually-banded archives (corals, shells, sclerosponges, tree wood, varved lake and marine sediments, and glacier ice), the minimum record duration for inclusion in the database is 30 years. For all other archives (speleothems, non-varved lake and marine sediments), records must have a minimum duration of 200 years and contain at least five data points during the CE.


### 2.3.2 Chronological constraints

The PAGES2k temperature database (PAGES 2k Consortium, 2017) was used as a guide for minimum chronological control criteria. Records from annually-banded archives must be either cross-dated or layer-counted; records from non-annually banded archives must have at least one age control point near both the oldest and youngest portions of the record, with one

additional age control point somewhere near the middle required for records longer than 1,000 years.

### 2.3.3 Peer review and public availability

To qualify for inclusion in the database, isotope records must be published in a peer-reviewed journal (i.e., not university published theses and dissertations). Records included in version 1.0 of the database had to be published and publicly

available before 4 May 2018 (see definition in Section 2.2).

### 2.3.4 Ancillary data

In some cases, paired geochemical measurements are also included in the Iso2k database to complement interpretation of the isotopic data, such as paired trace elemental measurements (e.g., Sr/Ca or Mg/Ca) that accompany some carbonate $\delta^{18}O$

records from corals, sclerosponges, and planktonic foraminifera, or $\delta^{13}C$ data that accompany some carbonate records. Derived isotopic data for deuterium excess (dxs) are also included for glacier and ground ice, where paired measurements of $\delta^{18}O$ or $\delta^2H$ allowed the original authors to calculate this additional hydroclimatic indicator. Similarly, derived values for the $\delta^{18}O$ of seawater are available for coral and marine sediment records in cases where an independent temperature reconstruction was available for the same archive (e.g., Sr/Ca for corals and Mg/Ca for planktonic foraminifera). Where the

paired carbonate $\delta^{18}O$ and Sr/Ca or Mg/Ca records can be used to infer the $\delta^{18}O$ of seawater (Cahyarini et al., 2008; Elderfield and Ganssen, 2000; Gagan et al., 1998), both time series ($\delta^{18}O$ measured directly on carbonate and $\delta^{18}O$ seawater calculated from paired records) as well as the ancillary, non-isotopic geochemical records are included in the database (Section 4).

**2.4 Quality control procedure**

Records considered to be a primary time series for their respective sites (Section 4; Table 6) were quality controlled to the highest degree possible, as described below. Primary time series were judged to be the one or two time series upon which the



original authors based their main climatic interpretations. For archives such as corals and speleothems, the primary time series is typically a composite of multiple records from a site or the latest of a series of modified records from a site, whereas

for other archives the primary time series is one deemed to have the most robust climatic signal (e.g., for lake sediments, a biomarker of terrestrial versus mixed terrestrial/aquatic origin). Non-primary time series were quality controlled as much as possible and are included because they may contain valuable information for database users. Both data and required metadata fields were screened for accuracy and completeness by one or more project members, with initials of the project member performing the final quality control (QC) check included in the Iso2k_QC_certification metadata field. Records

were screened by their respective archive teams to ensure that criteria for inclusion in the database were met. Metadata fields that required standardized or controlled vocabularies were double checked to ensure those terms were adhered to (Section 4). During the quality control certification process, project members used a web-based data viewer (lipdverse.org) and other visualization tools to display the raw data and metadata.

Each metadata field in the database (Tables 1–7) has a quality control certification "level" from 1–3 , defined as follows:

- **Level 1** fields are required metadata for inclusion in the Iso2k database. These fields are generalizable enough to be suitable for all archive types, and they are recommended as primary fields for filtering, sorting, and querying records in the database. Level 1 required fields were subject to the highest QC standard. They follow standardized

Iso2k vocabularies, where appropriate (Table 7); geographical data were checked against maps, and interpretation fields were checked against the original publication. Examples of level 1 metadata include geographical (ISO 19115) and publication information (DOI), and the minimum required subset of isotope and proxy system interpretation metadata fields (see Section 4).

- **Level 2** fields are highly useful, but not required, metadata fields in the Iso2k database. They may be used as secondary fields for further filtering, sorting, and querying records in the database; these fields may be particularly useful for certain archives, or to refine interpretations after an analysis has been performed. Examples of Level 2 fields include species name (marine and lake sediments and corals) and compound chain length for compound-specific $\delta^2$H measurements (lake sediments). Terminology was standardized only where necessary and appropriate.

In other cases, these fields contain freeform text with direct quotes from the original publications. During the QC certification process these fields were checked against the original publication for clarity and consistency.

- **Level 3** fields may be useful to some users of the Iso2k database but are not generally recommended as fields for filtering and sorting records in the database. Level 3 fields are not entered as standardized vocabularies and the

information is sometimes not given in the original publications. Examples of level 3 fields include information pertaining to the integration time of a proxy sensor.





- **Automatic fields:** The database also contains several automatically-generated fields that were computed directly from the data records following QC certification. Fields use standardized vocabularies and units. Examples include

315         binary fields for whether the dataset contains raw chronological control data.

Ancillary data are not quality-controlled, but are included in LiPD format for reference.

## 3. Contents of Iso2k database

**3.1 Archive types within the Iso2k database**

The Iso2k database contains data from a variety of geological and biological archives. Following Proxy System terminology (Evans et al., 2013), each *archive* has one or more *sensors* that directly sense and incorporate *environmental* signals, i.e., the $\delta^{18}O$ and $\delta^{2}H$ of environmental waters, into their structures. Over time these sensors then form, are deposited into, or are otherwise imprinted upon an *archive* that is then subsampled and subjected to isotopic measurements or *observations*. In this

section, we describe the key characteristics of the archives and sensors that are important for the interpretation of the paleohydrological signals that they preserve.

Corals, sclerosponges, and mollusks

Corals, sclerosponges, and mollusks (predominantly bivalves and gastropods) form hard body parts of calcium carbonate

(aragonite or calcite) that record the conditions of the aquatic environment in which they live (see reviews of (Black et al., 2019; Corrège, 2006; Druffel, 1997; Evans et al., 2013; Sadler et al., 2014; Surge and Schöne, 2005). Further, except for sclerosponges (which are dated using U/Th geochronology), these aquatic carbonates contain annual banding structures, enabling precise chronology development. Reef-building corals represent the bulk of annually-resolved marine archives included in the Iso2k database. These corals are distributed in warm shallow waters throughout the tropical oceans, whereas

sclerosponges (i.e., coralline sponges or Demospongiae) and mollusks are found worldwide, the latter in both estuarine and freshwater environments. Micro-sampling and laser ablation technologies allow for sub-annual to annual sampling resolution in corals, mollusks, and sclerosponges for elemental (e.g., Sr/Ca, Mg/Ca) and isotopic analysis ($\delta^{18}O$ and $\delta^{13}C$). When living samples are collected in modern waters, they contain environmental archives of the recent past (decades to several centuries), whereas dead, fossil, and archaeological material can be radiometrically dated to provide windows of past

isotopic variability, some of which have been cross-dated with modern records (Black et al., 2019 and refs therein). The $\delta^{18}O$ signal in these archives represents a combination of linear, temperature-dependent isotopic fractionation, as well as changes in the isotopic composition of the surrounding water ($\delta^{18}O_w$) (Grottoli and Eakin, 2007; Rosenheim et al., 2005). In some regions, the temperature component dominates the $\delta^{18}O$ signal, whereas in other regions $\delta^{18}O_w$ variability is the primary driver of the $\delta^{18}O$ variability and reflects hydrological and/or oceanographic processes such as vertical and horizontal

advection or the freshwater endmember (Conroy et al., 2017; Russon et al., 2013; Stevenson et al., 2018). In many ocean




settings, the close coupling between ocean-atmosphere variability leads to co-occurring cool and dry (or warm and wet) anomalies that produce complementary isotopic anomalies (Carilli et al., 2014; Russon et al., 2013; Stevenson et al., 2015, 2018). In estuarine or freshwater settings, mollusk $\delta^{18}O$ values are closely linked to the local precipitation-evaporation budget (Azzoug et al., 2012; Carré et al., 2019). Coral $\delta^{18}O$ and $\delta^{13}C$ contain a vital effect and coral $\delta^{18}O$ is offset from

$\delta^{18}O_w$, whereas mollusk and sclerosponge $\delta^{18}O$ is generally precipitated in equilibrium with environmental water. Some coral $\delta^{18}O$ records in the Iso2k database have had their mean $\delta^{18}O$ removed by original authors for comparison and cross-dating with other coral records and this is noted in the metadata.

Glacier ice

Climate records from glacier ice are found primarily at high latitudes (Antarctica, Arctic) and high elevation (e.g., Andes, Himalayas) (Eichler et al., 2009; Meese et al., 1994). Glacier ice is formed from the accumulation of snow, which over time compacts into a section of chronologically continuous layered ice. Cores drilled through layers of glacier ice preserve sub-annually to centennially resolved climate information, with resolution varying among records due to snow accumulation rates and laboratory sampling and analysis methods (Rasmussen et al., 2014). Ice cores are dated through a variety of

methods; annual layer counting and alignment to volcanic horizons are the most common approaches for records spanning the CE (Sigl et al., 2014). This database contains records of $\delta^{18}O$, $\delta^2H$, and/or dxs of glacier ice. These proxies reflect the isotopic composition of precipitation (snowfall and ice), which is highly correlated to local temperature but additionally reflects changes in moisture source and condensation processes (Goursaud et al., 2019). Physical processes such as isotopic diffusion in the firn, melt and infiltration, and compaction of ice layers generally smooths the seasonal to interannual signal

of climate variability in glacier ice, and the potential influence of these processes is site specific.

Ground ice (wedge ice and syngenetic pore ice)

Ground ice includes all types of ice found in permafrost; wedge ice and syngenetic pore ice hold the largest potential for paleoclimate reconstructions (Opel et al., 2018; Porter et al., 2016). Ice wedges in permafrost landscapes form via repetitive

thermal contraction cracking in winter and infilling of frost cracks mostly by snowmelt in spring (with potential minor contribution of snow and/or depth hoar). The integrated isotopic composition of the previous winter's snow pack is transferred into a single ice vein without additional isotopic fractionation due to rapid freezing in the permafrost. Thus, ice wedges preserve precipitation of the meteorological winter and spring, with $\delta^{18}O$ and $\delta^2H$ commonly interpreted as proxies for local air temperature (Meyer et al., 2015). Ice-wedge records are temporally constrained by radiocarbon dating of

macrofossils or dissolved organic carbon in the ice. Conversely, pore ice in syngenetic permafrost integrates precipitation that reaches the maximum thaw depth in the late summer. The pore ice seasonality is a function of the local precipitation climatology and residence time of active layer pore waters, and pore ice is enriched in heavy isotopes relative to the initial pore waters due to equilibrium fractionation during freezing (O'Neil, 1968) . Because syngenetic pore ice formed within



accumulating surface sediments, its age can be modeled based on a radiometrically constrained sediment age-depth profile.

Syngenetic pore ice can be cored and sub-sampled in the same way as glacier ice (Porter et al., 2019).

Lake sediments

Lake sediments may provide long and continuous records of past environmental change (Dee et al., 2018; Mills et al., 2017), and preserve a number of sensors for oxygen and hydrogen isotopes (Leng and Marshall, 2004). Carbonate minerals—

precipitated inorganically from lake waters or in the shells of aquatic invertebrates—have been used as sensors for the isotopic composition of lake water (Hodell et al., 2001; Jones and Dee, 2018; Von Grafenstein et al., 1998). Additional proxies analysed with increasing frequency include biogenic silica (mostly from diatoms; e.g., (Chaplplin et al., 2016; Swann et al., 2018), cellulose (Heyng et al., 2014), chitinous invertebrate remains (Van Hardenbroek et al., 2018) and lipids (Konecky et al., 2019a; Sachse et al., 2012). Of these proxies, the oxygen isotope composition of carbonates and silicates is

subject to temperature-dependent isotope fractionation during mineralisation, whereas the isotopic composition of organic materials is generally not influenced by temperature (Rozanski et al., 2010). The compound-specific hydrogen isotopic composition of a lipid reflects the environment in which the organism producing the lipid grew. Lipids produced by aquatic macrophytes or algae reflect the isotopic composition of the lake water, whereas lipids produced by terrestrial plants reflect the isotopic composition of soil or leaf water (which is, in many cases, highly influenced by the isotopic composition of

precipitation). Both types of lipids are preserved in lake sediments (Castañeda and Schouten, 2011; Rach et al., 2017; Thomas et al., 2016).

For sensors that record the $\delta^{18}O$ or $\delta^2H$ of lake water, the climatic or hydrological change recorded in $\delta^{18}O$ or $\delta^2H$ depends primarily on the degree to which evaporation influences the lake's hydrological balance (Gibson et al., 2016). In turn, the

effect of evaporation on lake water isotopes largely depends on the residence time of water within the lake system, and the degree of hydrological 'closure' of the lake. In open lake systems—which often have surface water inflows and outflows, with a resulting short water residence time—lake waters often reflect the isotopic values of the inflowing waters, which itself generally approximates, a (sometimes) lagged, signal of the weighted mean of the isotopic composition of local precipitation (Jones et al., 2016; Tyler et al., 2007). In hydrologically closed lakes—often without surface outflows and where more water

leaves the system through evaporation—the initial isotopic composition of inflowing waters is altered due to this evaporation, with the $\delta^{18}O$ or $\delta^2H$ of water increasing with increasing evaporation (Dean et al., 2015; Leng and Marshall, 2004).

Wood

The wood in tree rings (tree-ring cellulose) is one of the few terrestrial proxy archives that can be directly constrained to calendar years (McCarroll and Loader, 2004; Schweingruber, 2012). Seasonal to annual information about climatic and environmental changes is recorded in tree-ring cellulose $\delta^{18}O$. The $\delta^{18}O$ of tree-ring cellulose is influenced by the $\delta^{18}O$ of



leaf water, which in turn depends upon the $\delta^{18}O$ of precipitation-derived soil water and its evaporative $^{18}O$-enrichment in the leaf as dictated by physiological traits and ambient humidity (Barbour et al., 2004; Roden et al., 2000). Equilibrium

biosynthetic fractionation causes cellulose precursors (e.g., glucose) to be enriched relative to the bulk leaf water by ~27‰ (Sternberg et al., 1986). As the biosynthetic fractionation is relatively constant (Cernusak et al., 2005), the environmental factors that influence the $\delta^{18}O$ of water used by plants during photosynthesis dictates the fluctuation of $\delta^{18}O$ of the tree-ring cellulose. The primary climatic signals vary widely by latitude and degree of continentality. For example, temperature typically influences cellulose $\delta^{18}O$ at mid- to high-latitude sites (e.g., (Churakova et al., 2019; Porter et al., 2014; Saurer et

al., 2002; Sidorova et al., 2012) whereas precipitation amount influences cellulose $\delta^{18}O$ in tropical or monsoon affected regions (e.g., (Brienen et al., 2013; Managave et al., 2011).

### Speleothems

Speleothems are secondary cave deposits that form when water percolates through carbonate bedrock. Both atmospheric

$CO_2$ and $CO_2$ generated by plant root respiration and organic matter decomposition are dissolved into rainwater as it percolates through the soil, producing carbonic acid that rapidly dissociates to produce weakly acidic water. As this acidic water percolates through the bedrock, it dissolves carbonate until the water becomes supersaturated with respect to calcium and bicarbonate (Fairchild and Baker, 2012). When the percolating waters emerge in a cave, $CO_2$ degassing from the drip water to the cave atmosphere induces $CaCO_3$ precipitation, resulting in the formation of stalagmites and stalactites (Atkinson

et al., 1978) that preserve the $\delta^{18}O$ signal of the waters that have percolated through from the surface (Lachniet, 2009). The $\delta^{18}O$ of the deposited carbonate therefore reflects the $\delta^{18}O$ of soil/groundwater that infiltrates, which is strongly influenced by the $\delta^{18}O$ of precipitation but with additional influences of aquifer mixing times, seasonality of infiltration, and in some cases extreme events (Moerman et al., 2014; Taylor et al., 2013) processes within the karst and cave, such as calcite precipitation prior to speleothem deposition and/or kinetic isotope effects, can alter the $\delta^{18}O$ of the deposited carbonate.


Although there are hydroclimatic limits on speleothem growth, speleothem distribution is largely constrained by the presence of carbonate bedrock (Fairchild and Baker, 2012). Speleothems form in a wide range of hydroclimate conditions, from extremely cold climates in Siberia to arid regions in the Middle East and Australia. The temporal resolution of speleothem paleoclimate series ranges from sub-annual to centennial, and primarily depends on the karst and cave

environment. Due to the high precision of uranium-series dating, speleothems provide opportunities to determine the timing of regional hydrological response to global events and links to external forcing mechanisms (e.g., insolation changes) (Fischer, 2016). The different types of measurements made on speleothems—including $\delta^{18}O$, $\delta^{13}C$, and various trace elements—can be used to reconstruct past changes in the hydrological cycle.

### Marine sediments





Marine sediments contain two types of sensors that have widely been used for measuring water isotope variability: planktonic foraminifera and biomarkers. Planktonic foraminifera are unicellular zooplankton living in the upper hundreds of meters of the ocean. They build a calcite skeleton, which is preserved in the sediment. The $\delta^{18}O$ of planktonic foraminifera calcite reflects a spatially (and temporally) variable combination of temperature and $\delta^{18}O_{sw}$ (Urey, 1948) and to a lesser
degree also the seawater carbonate ion concentration (Spero et al., 1997), although changes in the latter parameter are likely negligible during the CE. The temperature effect on the $\delta^{18}O$ of foraminifera calcite is systematic, the $\delta^{18}O_{sw}$ can be reconstructed using (species-specific) paleotemperature equations in conjunction with an independent estimate of calcification temperature based on Mg/Ca (Elderfield and Ganssen, 2000). Planktonic foraminifera have a short life cycle (about a month) and species-specific seasonal and depth habitat preferences (Jonkers and Kučera, 2015; Meilland et al.,
2019), such that any planktonic foraminifera record bears an imprint of the ecology of the sensor (Jonkers and Kučera, 2017).

Biomarkers in marine sediments are lipids synthesized either by marine photoautotrophs, which track past changes in surface seawater isotopic values, or from vascular plants, which track soil water isotopic values on an adjacent land mass (Sachse et
al., 2012). Biomarkers are strongly affected by isotopic fractionation during lipid biosynthesis, and that fractionation is often assumed to be constant (Sachse et al., 2012). However, as for planktonic foraminifera, biomarker $\delta^{2}H$ values are also affected by a combination of environmental parameters. The $\delta^{2}H$ values of $C_{37}$ alkenones (synthesized by coccolithophorids) are impacted by fractionation that changes with salinity and growth rates (Schouten et al., 2006), which can mask changes in the $\delta^{2}H$ of seawater. The sources of leaf waxes are terrestrial plants, and the processes affecting leaf waxes in marine
sediments are the same as in lake sediments but generally with longer associated time lags between the sensor recording the $\delta^{2}H$ of soil water and ultimate deposition in the marine sediment archive.

## 4. Description of Iso2k metadata fields

The Iso2k database contains over 180 metadata fields. The 55 main fields are described in Tables 1–6; 23 of these were
strictly quality-controlled following the Level 1 definition in Section 2.4. Entries for some required metadata fields were standardized with controlled vocabulary to allow users to easily query the database for records based on archive type, isotope ratio (O or H), waters from which the isotope ratios are derived, materials on which the isotope ratios were measured, or the environmental parameter that controls isotopic variability **[Figure 1]**. Metadata fields describe the primary isotopic variable being inferred, i.e., the 'isotope interpretation' (e.g., the $\delta^{2}H$ of precipitation), the water from which it was inferred, i.e.,
'inferred material' (e.g., soil water), the material that was actually measured, i.e., 'measured material' (e.g., long-chain *n*-alkane components of leaf waxes), and information about the original climate interpretation. Distinction between the archive type **[Figure 2]**, inferred material **[Figure 3]**, and the isotope interpretation **[Figure 4]** allow for advanced analyses and straightforward data-model comparisons using the database. These metadata interpretation fields were derived from interpretations reported in the original publications. Below and in Tables 1–6, we describe key metadata fields in the




database, including all Level 1 and Level 2 fields (see Section 2.4 for a description of levels). Table 7 provides standardized
vocabularies and common terminologies. Table 8 provides selected chronological control metadata. Supplementary Table 1
gives key metadata for each primary time series (Section 2.4), including all Level 1 fields and selected additional Level 2
fields, and references to original publications (citations also listed in Supplementary Tables 2 and 3).

**4.1 Entity metadata**

The entity metadata fields provide basic information for each record, including the isotope measured, the archive type,
location (longitude, latitude, and elevation), start and end dates of each record, and both the DOI and citation for the original
publication. Entries for *archiveType*, *paleoData_variableName*, and *paleoData_units* metadata fields are standardized
(Table 7) across all archive types to facilitate easy querying and analyses. Each record is assigned a unique LiPD identifier,
and all isotope records are assigned a unique Iso2k identifier. The alphanumeric Iso2k identifiers contain 11 characters and
digits as follows: archive type (2 characters), year published (2 digits), first author's last name (2 characters), site name (2
characters), sample number (e.g., 00, 01, 02, 03…) for different cores or core composites from the same site, and letter (A,
B, C…) for multiple time series derived from the same core. A list and detailed description of key entity metadata fields are
provided in Table 1.


The *paleoData_variableName* indicates the variable measured for each *archiveType*, usually $\delta^{18}O$ or $\delta^{2}H$. In some cases
other paired geochemical measurements are included in the database to complement interpretation of the isotopic data
(section 2.3.4).

**4.2 Paleodata metadata**

The paleodata metadata fields provide information for each proxy record; a detailed description of key paleodata metadata
fields are provided in Table 2. Measured and derived water isotope time series are identified using the
*paleoData_variableType* and *paleoData_description* fields, and should not be confused with the isotope interpretation
metadata fields (section 4.3), which more broadly refer to the way each proxy record is interpreted (e.g., speleothem
carbonate interpreted as a proxy for the $\delta^{18}O$ of precipitation). The variable description (*paleoData_description*) is the
general category of material that was measured for its isotopic ratio (e.g., carbonate or terrestrial biomarker). Further details
are given by *measurementMaterial*, which is a more specific description of what was measured (e.g., coral, glacier ice, lake
sediment), and *measurementMaterialDetail,* which provides further specificity of the *measurementMaterial*, such as mineral,
species, or compound. In contrast, the *inferredMaterial* field indicates the environmental source waters whose isotope
variability is inferred (e.g., precipitation, lake water, groundwater) **[Figure 1]**. The environmental source waters in the
*inferredMaterial* field are not meant to be highly specific (e.g., intracellular leaf water) but rather broad pools of
environmental waters that have direct analogs or counterparts in climate models.





**4.3 Isotope interpretation metadata**

The isotope interpretation metadata fields compile critical information about environmental variables that influence isotopic variability within each record (Table 3). These fields indicate the environmental variable thought to exert dominant control on isotopic variability of the inferred environmental source waters (*inferredMaterial*) of each record, the mathematical relationship between the isotope interpretation variable and the isotope record, and the season(s) during which this interpretation applies. All isotope interpretation fields in the database are prefaced by *isotopeInterpretation*. The

*isotopeInterpretation1_variable* field lists the primary driver of isotopic variability in the environmental source waters according to the original publications, for example air temperature or relative humidity (Table 7). For records where multiple variables can explain some fraction of the variability, the *isotopeInterpretation2* and *isotopeInterpretation3* fields are also populated. The *isotopeInterpretation1_direction* is a field that gives the sign (positive or negative) of the relationship between the isotope measurements and the environmental variable.


The *isotopeInterpretation1_variableGroup* field is a simplified supergrouping of terms in the *isotopeInterpretation1_variable* field in order to facilitate comparisons across different archives and realms, with three options (temperature, isotopic composition of precipitation ('P_isotope'), or effective moisture). Controlled vocabulary for metadata fields *isotopeInterpretation1_variable* and *isotopeInterpretation1_variableGroup* are standardized across all

archive types (Table 7).

The isotope interpretation metadata fields reflect the isotope systematics of the environmental source waters, and as such are distinct from the climatic inferences that one can make from a proxy record (Section 4.4). In some publications, this distinction is explicitly spelled out. For example, the cave drip water that becomes incorporated into the $\delta^{18}O$ of speleothem

carbonate in Borneo reflects the $\delta^{18}O$ of water mixed throughout an aquifer system over many months, which ultimately reflects a smoothed version of precipitation $\delta^{18}O$ (Moerman et al., 2014). In that case, the *inferredMaterial* is soil/groundwater and the *isotopeInterpretation1_variable* is $\delta^{18}O_{precipitation}$ ('P_isotope'). Separately, $\delta^{18}O_{precipitation}$ at that same study site reflects multiple hydroclimatic processes such as moisture transport and precipitation amount that lend it a regional imprint of the El Niño Southern Oscillation (ENSO) (Moerman et al., 2013), and so the climate interpretation of

speleothem $\delta^{18}O$ is related to ENSO, which would be described separately in the climate interpretation fields (Section 4.4). In many publications, the isotope systematics of the environmental source waters and the climate interpretation are stated implicitly rather than explicitly (e.g., by stating that the $\delta^{18}O$ of speleothem carbonate reflects monsoon intensity, or by stating that it reflects local precipitation amount via the amount effect (Dansgaard, 1964). In these cases, the *isotopeInterpretation1_variable* is still 'P_isotope' and information about the climatic interpretation is included in the

climate interpretation fields. These distinctions are critical for facilitating comparisons with isotope-enabled climate models, where complex and nonstationary climate/isotope relationships can be examined directly.



For *isotopeInterpretation1_seasonality,* some proxy sensors and/or archives are interpreted to record a seasonally-biased signal whereas others may record climate at an annual or sub-annual resolution (e.g., corals, some speleothems, 550    sclerosponges, mollusks, wood). If the record is interpreted to be biased towards a specific season, the calendar months corresponding to that season—given as the first letter of each month unless clarification is necessary—are recorded in the metadata field (e.g., MAM, DJFM, Jan). If the record represents an approximately mean annual signal, 'annual' is recorded in the seasonality field. For coral records, if the record has sub-annual resolution (e.g., sampled at monthly or bimonthly intervals) but the overall record is not biased to any particular season, 'sub-annual' is recorded in the metadata field.


### 4.4 Climate interpretation metadata

In contrast to the isotope interpretation (Table 3), climate interpretation metadata (Table 4) represent the original authors' expert judgment about the primary climatic controls on the isotope ratios at their study site. Climate interpretation metadata specify either climatic variables (e.g., temperature, precipitation amount) or processes (e.g., the Pacific Decadal Oscillation, 560    Asian monsoon intensity) that the authors interpreted to influence the isotopic composition of the proxy record, and as such, they are neither standardized nor quality controlled. These metadata are included as useful background information, but should not serve as a primary filter for users of the Iso2k database. A user might filter records based on the isotope interpretation field, then check the climate interpretation field for a qualitative understanding of which climatic processes may be important for the filtered set of records. For records where the *isotopeInterpretation2* and *isotopeInterpretation3* 565    metadata are populated (Table 3), the corresponding *climateInterpretation2* and *climateInterpretation3* metadata may also be provided.

### 4.5 Queryable and standardized metadata

To make the database more user-friendly and queryable, some metadata fields contain logical flags (e.g., 0 or 1, true or 570    false), cross-links (e.g., to a corresponding record ID in another PAGES2k database), or geographic labels (e.g., continent or ocean basin) that allow for easy sorting (Table 6). For example, if a record was included in the PAGES2k temperature database and reconstructions (Abram et al., 2016; Kaufman, 2014; PAGES 2k Consortium, 2017; Stenni et al., 2017; Tierney et al., 2015), that record is cross-linked to its associated PAGES2k ID wherever possible, permitting easy database query and analysis of records in only one database and those common to both databases. Approximately 15% of the records in the 575    Iso2k database were also incorporated into other PAGES2k compilations with the most overlap occurring in coral records and high-latitude ice cores. For these records, the extensive metadata can be used to facilitate deeper analyses of the hydroclimatic signals contained in these mainly temperature-dominated isotopic records. For example, with coral $\delta^{18}$O records, many of which are included in both the PAGES2k temperature and Iso2k databases, the isotope interpretation fields denote the relative influence of $\delta^{18}O_{sw}$ vs. temperature to the isotopic variability of the coral carbonate skeleton.

580



### 4.6 Chronological control data

Chronological or depth-age metadata provides essential chronological information for isotope records across all archive types, including an age model and the average temporal resolution for each isotope record. For non-annually banded records, age-depth models and radiometric dating information (Table 8) are included where available to facilitate independent age modeling. This information is stored in 'chronData' tables that are linked to the measured data ('paleoData') tables. If there are raw chronology data associated with a record (e.g., radiometric age determinations), *hasChron* is set to 1; otherwise this parameter is 0. Similarly, if sample depth data are available (e.g., core depth), *hasPaleoDepth* is set to 1.

To support the information implicit within each record's age-depth model, chronological metadata are provided for all individual age constraints (when available) and these metadata are summarized in Table 8. If available, sample information (*thickness* and *labID*) is provided for all age constraints. Each age constraint that is not in radiocarbon years has *age* in calendar years before 1950 CE, and *ageUncertainty*. Radiocarbon age constraints have *age14C* in radiocarbon years before 1950 CE and *age14Cuncertainty*. The *materialDated*, *reservoirAge14C*, and *reservoirAge14Cuncertainty* are also provided for radiocarbon age constraints to allow users to derive their own age-depth models if desired. For radiocarbon ages, we also provide *fractionModern*, *fractionModernUncertainty*, *delta13C* (of the material that was radiocarbon dated), and *delta13Cuncertainty* when available.

Several lake and marine sediment archives contain measurements of radiogenic isotopes—$^{210}$Pb, $^{137}$Cs, and/or $^{239+240}$Pu—to constrain the age of the sediment at and near the surface/core top. Where applicable, we provide the isotope *activity* and the *activityUncertainty*. For $^{210}$Pb measurements, the *supportedActivity* field is Y if the activity is supported by $^{210}$Pb production in the surrounding matrix and N if the activity is not supported. The *x210PbModel* describes the type of model used to determine the age based on the radiogenic isotope measurements. For carbonate systems such as speleothems and corals, U/Th dating is often used. Where available, chronological tables in the database contain information about the $^{238}$U and $^{232}$Th content (*U238*, *Th232*), the $^{230}$Th/$^{238}$U activity ratio (*Th230_U238activity*), $\delta^{234}$U (*d234U*), and their uncertainties (*U_Thactivity_error* and *d234U_error*). Fields such as the initial $^{234}$U/$^{238}$U (*dU234intial*) and $^{230}$Th/$^{232}$Th activity ratios (*Th230_Th232ratio*) are also included for correcting ages for the initial $^{234}$U/$^{238}$U activity, and detrital thorium contamination, respectively.

The *useInAgeModel* is a binary field where Y indicates that age constraint was used in the published age model and N indicates that age constraint was not used in the published age model.

The amount and type of uncertainty in each chronology are provided in *paleoData_chronologyIntegrationTimeUncertainty* and *paleoData_chronologyIntegrationTimeUncertaintyType* respectively, while *paleoData_chronologyIntegrationTimeBasis* outlines how the chronology was constructed. By contrast, the



*paleoData_sensorIntegrationTime*, *paleoData_sensorIntegrationTimeBasis*, *paleoData_sensorintegrationTimeUncertainty*, *paleoData_sensorIntegrationTimeUncertaintyType*, and *paleoData_sensorIntegrationTimeUnits* fields—where available— describe the amount of time over which a sample integrates isotopic values.

## 5. Key characteristics of Iso2k data records

**5.1 Spatial, temporal, archival, and isotopic characteristics of data coverage**

The Iso2k database contains 756 stable isotope ($\delta^{18}O$, $\delta^2H$) records from 505 unique sites. There are 10 archive types, including: 145 records from annually-banded skeletal carbonate marine archives (corals (n = 140), sclerosponges (n = 4), and mollusks (n = 1)); 205 from glacier ice (n = 200) and ground ice (n = 5); 158 from lake or terrestrial sediments, 99 from marine sediments, 68 from speleothems, and 81 from wood **[Figure 2a]**. The database is primarily composed of $\delta^{18}O$ records

(65.1%) and $\delta^2H$ (9.7%), with 12 sites having records of both isotope systems (derived from the same sensor in ice cores, or different sensors in lake sediments). 255 additional records containing ancillary data (e.g., $\delta^{13}C$, Mg/Ca, Sr/Ca) are also included. Of the 756 records, 601 are considered 'primary' $\delta^{18}O$ or $\delta^2H$ time series (Supplementary Table 1 and Section 2.4), including 101 records from annually-banded skeletal carbonate marine archives (corals (n = 96), sclerosponges (n = 4), and mollusks (n = 1)), 165 from glacier ice (n = 161) and ground ice (n = 4), 114 from lake or terrestrial sediments, 95 from

marine sediments, 47 from speleothems, and 79 from wood.

Spatial coverage of the sites in the database is global, but most sites are from the low latitudes and Northern Hemisphere mid-latitudes **[Figure 2a; Figure 4b]**. Data availability is low for most of the Southern Hemisphere, with the exception of glacier ice records from Antarctica **[Figure 4b]**. The temporal coverage increases from about 250 proxy time series near the

year 0 CE to more than 400 time series at the beginning of the twentieth century **[Figure 2b]**. The average length and resolution of each $\delta^{18}O$ time series vary considerably and are archive-dependent. Banded, biologically-derived archives (corals, sclerosponges, mollusks, and wood) offer the highest resolution (monthly to seasonal), and a temporal extent of between 24 years to 375 years for corals and 38 to 1030 years for tree records (timespan is the 2.5–97.5% quantiles). Layer-counted archives such as glacier ice generally offer annual resolution and a time span between 41–1979 years. Other

archives have lower resolution, but provide more continuous coverage across the CE. The median resolution of records is 12 years/sample for speleothems, 25 years/sample for lake sediments, 28 years/sample for marine sediments, and 97 years/sample for ground ice, and the median time span of records in these archives is >1200 years. These lower resolution time series almost exclusively make up the records in the database prior to ~1700 CE and prevent a drop in coverage in older time periods described in other PAGES2k compilations (PAGES 2k Consortium, 2013).


The records in the Iso2k Database capture many aspects of hydroclimate **[Figure 4]**. The first-order interpretation (*isotopeInterpretation1_variable*) for 44% of the $\delta^{18}O$ and $\delta^2H$ records in the database is 'P_isotope', meaning that $\delta^{18}O$ and $\delta^2H$ of the inferred material (ice, soil water, seawater, etc.) is primarily driven by the $\delta^{18}O$ and $\delta^2H$ of precipitation. The first-




order interpretation for 26% of the records in the database is 'T_water' or 'T_air', meaning that the temperature of water or

air is the primary driver of $\delta^{18}O$ and $\delta^2H$ of the inferred material. Finally, 24% of records in the database are primarily driven

by some aspect of evaporation or evapotranspiration, collectively referred to as 'Effective Moisture' in the

*isotopeInterpretation1_variableGroup* category. This category includes 'd18O_seawater' (driven by ocean circulation and

by precipitation/evaporation at the sea surface) , 'ET' (evapo-transpiration), 'I_E' (infiltration/evaporation), and 'P_E'

(precipitation/evaporation) entries for *isotopeinterpretation1_variable*.


**5.2 Validation**

There is currently no existing observational dataset of isotope ratios in all major pools of the water cycle that can serve as a

true validation of the Iso2k database. However, the vast majority of ice records in the Iso2k database have an inferred

material of 'precipitation' and a first-order isotope interpretation of 'P_isotope'. For these records, the $\delta^{18}O$ averaged for the

twentieth century (all data points after 1900 CE) provides a reasonable match with the observed annual average $\delta^{18}O$ of

precipitation from the Global Network of Isotopes in Precipitation (GNIP) (Terzer et al., 2013) **[Figure 5]**. This provides

confidence that the isotopic data contained in the Iso2k database can reasonably be used for analyses such as calculation of

latitudinal gradients in $\delta^{18}O$ over the CE, even before accounting for seasonal biases and other transformations within the

proxy system. We note that while other proxy data types such as speleothems and leaf wax biomarkers are sensitive to

P_isotope (and *isotopeinterpretation1_variable* for many of these records is listed as 'P_isotope'; Figure 4), their most direct

inferred materials are meteoric waters such as soil water or groundwater rather than precipitation and are therefore not as

directly comparable to the GNIP database.

**6. Usage notes**

**6.1 General applications**

The Iso2k database is the most comprehensive database of paleo-water isotope records to date for the CE. For the first time,

this database allows investigation of spatial and temporal hydroclimate variability from regional to global scales across

multiple proxy systems. Using the 'inferred material' metadata, the database can be directly compared with the output of

climate models, allowing investigation of the water cycle in far greater depth than was previously possible.


Alongside the data itself, the detailed 'isotope interpretation' metadata fields are the foundation of this database. These fields

allow users to understand the processes reflected in the isotope data, and filter the database according to particular scientific

questions. For example, a user may be interested in the temporal variability of isotope records driven primarily by changes in

effective moisture, and the Iso2k standardized vocabulary means that it is straightforward to filter for these records. Note

that for many records in the database, isotopic variability is affected by more than one variable and these secondary

influences may not be trivial when conducting meta-analyses. Although only '*isotopeinterpretation1*' fields have been





quality-controlled to the highest level, the subsequent isotope interpretation fields also contain well-curated information that is important for data interpretation.

### 6.2 Example workflow for filtering and querying data records


Records in the Iso2k database are provided as published (i.e., not re-calibrated or validated). This preserves the large amount of information contained within water isotope proxy measurements that would be lost if condensed to reconstruct discrete variables. Rather, we leave it to the database users to filter and assess records as needed.

For initial querying of the database, in nearly all cases, we recommend first filtering by the following:

1. variableName = 'd18O' or 'd2H' (excludes any non-isotopic data)
2. paleoData_units = 'permil' (excludes records published as z-scores or anomalies)
3. paleoData_iso2kPrimaryTimeseries = 'TRUE' (includes only primary time series for each site)

Additional filtering of records should be performed using Level 1 or Level 2 fields. For example:

- isotopeInterpretation1_variable = 'P_isotope' (includes only records where the first-order control of isotopic variability is the isotopic composition of precipitation)
- paleoData_description = 'carbonate' or 'terrestrial biomarker' or 'tree ring cellulose' (to extract terrestrial archives sensitive to P_isotope aside from ice cores), or:

- paleoData_inferredMaterial = 'groundwater' or 'soil water' or 'lake water' (accomplishes similar results to the above)

Additional filtering of records may be useful with other Level 2 fields, for example:

- climateInterpretation1_variable = contains 'P' or 'Precipitation_amount' or 'P_amount' (to extract only records

where authors' primary climatic interpretation was based on the amount effect)

The sample code provided with this dataset (Supplementary Material) provides a similar example to users.

### 6.3 Versioning scheme

This publication marks Version 1.0.0 of the Iso2k database (editors and reviewers: please note that you are reviewing version 0.14.2; this will be become version 1.0.0 upon publication, following any edits during the review process). Following publication, the database will continue to evolve, as new datasets are added (both new studies and previous records that have been missed) and existing data or metadata are extended, or as necessary, corrected. Readers who know of missing datasets, or who find errors in this version are asked to contact one of the lead authors, or submit new or edited

datasets directly through http://lipd.net/playground. As the database updates, it will be versioned following the scheme used





by other PAGES data collections (Kaufman et al., in revision; McKay and Kaufman, 2014; PAGES 2k Consortium, 2013, 2017), with the following format: X1.X2.X3, where X1, X2 and X3 are incrementing integers. When X1 increases, X2 and X3 reset to zero. When X2 increases, X3 resets to zero. X1 represents the number of publications describing the database. X2 increments each time the set of records in the database changes (addition or removal of a dataset). X3 increments when

the data or metadata within the dataset change, but the set of records remains the same. Upon updates, extensions or corrections to the database, rather than issuing errata to this publication, changes will be included in subsequent versions of the database and updated and described through the online data repository.

### 6.4 Availability of data and code

Following the previous PAGES2k and the Temperature 12k data compilations (Kaufman et al., in revision; PAGES 2k Consortium, 2017), the Iso2k database employs the Linked Paleo Data (LiPD) format (McKay and Emile-Geay, 2016), with serializations available for R, MATLAB, and Python. The LiPD format is machine-readable, with codebases to facilitate input, output, visualization and data manipulation in R, Python and Matlab. Simple visualization and data access (both as LiPD and csv files) is available through the LiPDverse at http://lipdverse.org/iso2k/current_version/. The LiPDverse

additionally houses other paleoclimate records and compilations that may be of interest to users of the Iso2k database. The serializations contain all LiPD files included in the current version of the Iso2k database. Serializations of the database can be downloaded from https://doi.org/10.6084/m9.figshare.11553162 (McKay and Konecky, 2020).

### 6.5 Citation

We encourage users of the database to not only cite the Iso2k data product but also the original publications and primary data sources (Supplementary Tables 2 and 3), particularly when analyses make explicit use of individual records.

### 7. Conclusions and anticipated applications of the Iso2k database

The global extent, quantity and quality of metadata included in the Iso2k database allow examination of the multiple
variables that impact water isotopes, including moisture source and transport history, temperature, and precipitation amount. These multivariate controls mean that water isotopes contain a wealth of information about climate. Importantly, water isotope signals contained in proxy archives can be modified by local environmental processes such as evaporation, biosynthetic fractionation, bioturbation in sediments, or diffusion. These archive- or proxy-specific transformations therefore additionally allow for reconstruction of water balance (E:P), different forms of drought (e.g., meteorological, hydrological or
soil moisture), and relative humidity (Rach et al., 2017). It is difficult to tease apart the effects of multiple variables in a single proxy record, but this global compilation of water isotope proxy records from a range of archives will help to overcome this barrier, facilitating extraction of common signals from the noise of individual proxies, and providing insights into different aspects of the hydrological cycle at a range of spatial and temporal scales.





The Iso2k database also provides an unprecedentedly direct comparison for state-of-the-art water isotope-enabled climate models. Many data-model comparison efforts compare climate model variables such as temperature and precipitation to paleoclimate data; the latter is often a complex and nonlinear signal integration of multiple climate influences, and uncertainties arise from the assumptions that must be made (Dee et al., 2016). Comparing water isotope fields from climate model outputs to isotope proxy records of the same components of the water cycle circumvents these uncertainties,

providing a more direct comparison of proxies and model simulations in the same units. Model validation on this relatively level playing field will improve estimates of climate models' ability to simulate changes in hydroclimate on long timescales. For those archives that further filter the isotopic signal, proxy system models can aid data model comparison (Dee et al., 2015, 2018; Jones and Dee, 2018). Therefore, the Iso2k database will not only enable global-scale comparisons with isotope-enabled climate models, but may also serve as an input database for paleoclimate data assimilation reconstructions such as

the Last Millennium Reanalysis (Hakim et al., 2016; Steiger et al., 2014) and the Paleo Hydrodynamics Data Assimilation (Steiger et al., 2018).



**Tables**

**Table 1: Key entity metadata (\*bold = Level 1 or required fields in database, *italics are references to other metadata or variable in the database*)**

| Variable | Name of field in database | Additional description | QC Level |
|---|---|---|---|
| Archive type | **\*archiveType** | Type of proxy archive (Table 2 and Table 7). | 1 |
| Latitude | **\*geo_latitude** | Site latitude in decimal degrees (-90 to +90). | 1 |
| Longitude | **\*geo_longitude** | Longitude in decimal degrees (-180 to +180). | 1 |
| Elevation | **\*geo_elevation** | Site elevation in meters relative to mean sea level (- below sea level, + above sea level). | 1 |
| Site name | **\*geo_siteName** | Name of the site, locality of nearest geopolitical center/municipality if applicable (i.e., islands retain their names). | 1 |
| Dataset ID | **\*dataSetName** | Iso2k-specific identifier assigned to all isotope records from a given site and publication. | 1 |
| Unique record ID | **•paleoData_iso2kUI** | Unique Iso2k identifier assigned to each isotope record to distinguish among records when more than one record exists in the original publication. | 1 |
| LiPD ID | **\*paleoData_TSid** | Unique LiPD file identifier for each time series in the database. | 1 |
| Variable name | **\*paleoData_variableName** | Variable measured (e.g., $\delta^{18}O$, $\delta^{2}H$). See Table 2 for more metadata and Table 7. | 1 |
| Variable units | **\*paleoData_units** | Units for *paleoData_variableName* (e.g., permil). See Table 2 for more metadata and Table 7. | 1 |
| LiPD link | **\*lipdverseLink** | Link to LiPDverse webpage. | 1 |
| Maximum year | maxYear | Maximum (most recent) year of each isotope record in calendar year (CE). See Table 8 for more chronology metadata. | auto |
| Minimum year | minYear | Minimum (earliest) date of each isotope record in calendar year (CE). See Table 8 for more chronology metadata. | auto |





| Publication DOI | pub1_doi | Digital Object Identifier for the first publication presenting the isotope record. | 1 |
| Publication citation | pub1_citation | Citation for the first publication presenting the isotope record. | 3 |
| Dataset DOI | datasetDOI | Digital object identifier for dataset assigned by original authors if available. | 3 |
| Dataset URL | paleoData_WDSPaleoUrl | URL linking back to records obtained from the NOAA NCEI data repository | 3 |



**Table 2: Key paleodata metadata (\*bold = Level 1 or required fields in database, *italics are references to other metadata or variable*)**


| Variable | Name of field in database | Description | QC Level |
|---|---|---|---|
| Variable description | **\*paleoData_description** | Human-readable description of *paleodata_variableName* (e.g., carbonate, $\delta^{18}O$ of glacier ice). | 1 |
| Measurement material | **\*paleoData_measurementMaterial** | Type of material in which *paleodata_variableName* was measured (e.g., coral, cellulose, biomarkers). | 1 |
| Measurement material detail | paleoData_measurementMaterialDetail | Free-form text with additional information about *paleoData_measurementMaterial*. | 2 |
| Inferred material | **\*paleoData_inferredMaterial** | Source water whose isotope variability is inferred (e.g., surface seawater, lake water, precipitation). See Table 7. | 1 |
| Inferred material group | **\*paleoData_inferredMaterialGroup** | Supergroup of inferred material, see Table 7 for controlled vocabulary. See Table 7. | 1 |
| Archive genus | paleoData_archiveGenus | Genus name of the archive, if available. | 3 |
| Archive species | paleoData_archiveSpecies | Species name of the archive, if available. | 3 |
| Values (data field) | paleoData_values | Field containing isotope time series or other measurements for each paleorecord. | 3 |
| Analytical uncertainty | paleoData_uncertaintyAnalytical | Analytical uncertainty in the measured variable when provided by the original publication; based on long-term precision of an internal standard of known value. | 3 |
| Analytical reproducibility | paleoData_uncertaintyReproducibility | Analytical reproducibility in the measured variable when provided by the original publication; based on repeat measurements of replicate samples, transects or cores from the same site. | 3 |
| Equilibrium evidence | paleoData_equilibriumEvidence | Indicates whether equilibrium conditions were present when the archive formed. | 2 |



| | | | |
|---|---|---|---|
| | | Indicates whether the isotope value was measured directly, temporally interpolated (e.g., from age tie points for annually-banded archives), or inferred (e.g., seawater isotopic variability, inferred from paired $\delta^{18}O$ and Sr/Ca or $\delta^{18}O$ and Mg/Ca in marine sediments). This information is also incorporated into paleoData_description. | 3 |
| Variable type | paleoData_variableType | | |



**Table 3: Key isotope interpretation metadata (*bold = Level 1 or required fields in database, *italics are references to other metadata or variable*)**

| **Variable** | **Name of field in database** | **Description** | **QC Level** |
|---|---|---|---|
| Primary isotope interpretation | **\*isotopeInterpretation1_variable** | Variable that controls isotopic variability within the record (e.g., 'Temperature_air', 'd18O seawater'). See Table 7. | 1 |
| Direction of relationship | **\*isotopeInterpretation1_direction** | Sign ('positive' or 'negative') of the relationship between the isotope values and the isotope interpretation variable. For example, a record with a temperature interpretation may have a decrease in $\delta^{18}O$, that corresponds to an increase in temperature. | 1 |
| Interpretation group | **\*isotopeInterpretation1_variableGroup** | Supergroup of isotope interpretations (one of temperature, effective moisture, or precipitation isotope ratio). See Table 7. | 1 |
| Mathematical relation | isotopeInterpretation1_mathematicalRelation | Type of relationship between isotope and climate variable ('linear' or 'nonlinear'). | 2 |
| Seasonality | isotopeInterpretation1_seasonality | The calendar months the isotope interpretation applies to is given as first initial of the months or as 'annual' or 'sub-annual' where applicable (e.g., corals, speleothems). | 2 |
| Basis | isotopeInterpretation1_basis | Basis for the isotope interpretation of each record as stated in the original publication (text or citation maybe given). | 2 |
| Coefficient | isotopeInterpretation1_coefficient | Numerical coefficient with interpretation variable. | 2 |
| Fraction | isotopeinterpretation1_fraction | Fraction of variance of explained by given climate variable. | 2 |




**Table 4: Key climate interpretation metadata**

| Variable | Name of field in database | Description | QC Level |
|---|---|---|---|
| Primary climate interpretation | climateInterpretation1_variable | Climate variables interpreted in each record (queryable freeform text with quotes from original publications; e.g., 'salinity', 'temperature'). | 2 |
| Primary climate interpretation detail | climateInterpretation1_variableDetail | Provides more information about the climate variable (e.g., sea surface for temperature or salinity). | 2 |
| Climate interpretation relationship direction | climateInterpretation1_direction | Sign ('positive' or 'negative') of the relationship between the isotope ratios and climate variable. For example, a record with a temperature interpretation may have a decrease in $\delta^{18}O$, that corresponds to an increase in temperature. | 2 |
| Climate interpretation basis | climateInterpretation1_basis | Basis for climate interpretation of each record as stated in the original publication. | 2 |



**Table 5: Key depth-age metadata (*bold = Level 1/required fields in database)**

| Variable | Name of field in database | Description | QC Level |
|---|---|---|---|
| Year (data field) | **\*year** | Field containing year data (units are CE) for the paleorecord. | 1 |
| Year units | **\*yearUnits** | Units of year data (CE). | 1 |
| Depth (data field) | depth | Depth in archive (e.g., in sediment core, stalagmite). | 2 |
| Depth units | depthUnits | Units of depth measurements. | 2 |
| Chronological integration time | paleoData_chronologyIntegrationTime | Average temporal resolution of each record in years/measurement. | 3 |
| Chronological integration time units | paleoData_chronologyIntegrationTimeUnits | Units for the *paleoData_chronologyIntegrationTime* field. | 3 |






**Table 6: Selected queryable metadata (*bold = Level 1/required fields in database)**

| Variable | Name of field in database | Description | QC Level |
|---|---|---|---|
| Has chronology? | hasChron | Indicates whether chronology data for the isotope record are available in the database. | auto |
| Record included in previous PAGES2k compilation? | paleoData_inCompilation | Indicates whether the record was used in earlier PAGES2k databases. | 2 |
| Ocean2k ID | paleoData_ocean2kID | Ocean2k unique ID for records included in both databases. | 2 |
| PAGES2k Dataset ID | paleoData_pages2kID | PAGES2k temperature dataset ID for records included in both databases. | 2 |
| QC Certification | **\*paleoData_iso2kCertification** | Initials of Iso2k Project Member that QC'ed the record. | 1 |
| Iso2k primary time series for dataset | **\*paleoData_iso2kPrimaryTimeseries** | For sites with multiple time series (e.g., caves with multiple stalagmites and a final composite), this time series should be primarily used ('TRUE' or 'FALSE'). | 1 |
| PAGES2k region | geo_pages2kRegion | The continental (e.g., 'SAm' for South America) or ocean (i.e., Ocean) regions corresponding to the PAGES2k or Ocean2k temperature reconstructions for the records included in those data compilations. | 3 |
| Ocean region | geo_ocean | The ocean region (e.g., Pacific) corresponding to the record site. | 3 |



**Table 7: Standardized controlled vocabulary options for metadata fields in the Iso2k database (Standardized labels show labels used in Iso2k Database, parentheses expand any abbreviations)**

| Metadata Field | Standardized labels |
|---|---|
| archiveType | coral, glacier_ice, ground_ice, lake_sediment, marine_sediment, mollusk_shells, terrestrial_sediment, speleothem, sclerosponge, wood |
| paleoData_variableName | d2H, d18O |
| paleoData_units | permil, zscore, permil_anomaly (specify relative to), PC (principal component) |
| isotopeInterpretation1_direction | positive, negative |
| isotopeInterpretation1_variable | T_water, d18O_seawater, P_E (precipitation/evaporation), I_E (input/evaporation), P_isotope, T_air, relative humidity, Veg (vegetation dynamics), ET (evapotranspiration: soilwater) |
| isotopeInterpretation1_variableGroup | - Temperature (comprising T_water, T_air)<br>- EffectiveMoisture (comprising d18O_seawater, P_E, I_E, relative humidity, Veg, ET)<br>- P_isotope |
| isotopeInterpretation1_inferredMaterial | Surface seawater (1 thermocline), subsurface seater, precipitation, lake water, soil water, lagoon water, groundwater |
| paleoData_inferredMaterialGroup | - Surface water (comprising surface seawater, lake water, lagoon water, subsurface seawater)<br>- Precipitation<br>- Soil/leaf water (comprising soil water, groundwater) |
| paleoData_measurementMaterial (*Level 2 QCed, not fully standardized*) | Coral, mollusk, ostracod, gastropod, glacier ice, aquatic or terrestrial biomarkers (n-alkane, n-alkanoic acid, dinosterol, botryococcene), planktonic foraminifera, cellulose, carbonate, or bulk carbonate |



**Table 8: Key chronological metadata**

| Variable | Name of field in database | Description |
|---|---|---|
| Age | age | Age in calendar years before 1950 CE (after any dating technique-specific corrections have been applied). |
| Age Uncertainty | ageUncertainty | 1 standard deviation uncertainty of calendar age. |
| Radiocarbon Age | age14C | Age in radiocarbon years before 1950 CE. |
| Radiocarbon Age Uncertainty | age14Cuncertainty | One standard deviation uncertainty of radiocarbon age in years. |
| Fraction modern $^{14}$C activity | fractionModern | Fraction of modern radiocarbon activity. |
| Fraction modern $^{14}$C activity uncertainty | fractionModernUncertainty | One standard deviation uncertainty of fraction of modern radiocarbon activity. |
| $\delta^{13}$C | delta13C | $\delta^{13}$C of material analyzed for radiocarbon. |
| $\delta^{13}$C uncertainty | delta13Cuncertainty | One standard deviation uncertainty of $\delta^{13}$C of material analyzed for radiocarbon. |
| Thickness | thickness | Thickness of the layer analyzed for the age constraint. |
| Lab Identifier | labID | Unique identifier provided by lab where age analysis was conducted. |
| Material Dated | materialDated | For radiocarbon age constraints, the material dated. |
| Activity | activity | $^{210}$Pb, $^{239+240}$Pu or $^{137}$Cs activity. |
| Activity Uncertainty | activityUncertainty | $^{210}$Pb, $^{239+240}$Pu or $^{137}$Cs activity uncertainty. |
| Supported Activity | supportedActivity | "Y" if supported $^{210}$Pb activity, "N" if unsupported $^{210}$Pb activity. |
| $^{210}$Pb model | x210PbModel | Model used to convert $^{210}$Pb activity to age (e.g., constant rate of supply). |
| $^{14}$C reservoir age | reservoirAge14C | $^{14}$C reservoir age. |
| $^{14}$C reservoir age uncertainty | reservoirAge14CUncertainty | $^{14}$C reservoir age uncertainty. |
| U/Th depth | depthUTh | Mid-point depth of the sub-sample drilled for U-Th age. |




| U/Th sample ID | sampleID | Sample ID for the U-Th age measured. |
|---|---|---|
| U/Th sample weight | weight | Weight of powder analyzed for U-Th age in mg. |
| $^{238}$U content | U238 | $^{238}$U content of the sub-sample in ppb. |
| $^{238}$U error | U238_error | Analytical uncertainty of $^{238}$U in ppb. |
| $^{232}$Th content | Th232 | $^{232}$Th content of the sub-sample in ppt. |
| $^{232}$Th error | Th232_error | Analytical uncertainty of $^{232}$Th in ppt. |
| d234U ratio | d234U | d234U ratio measured in the subsample. |
| d234U error | d234U_error | Analytical uncertainty of d234U. |
| $^{230}$Th/$^{238}$U activity | Th230_U238activity | [$^{230}$Th/$^{238}$U] activity measured in the subsample. |
| $^{230}$Th/$^{238}$U activity error | U_Thactivity_error | Analytical uncertainty of $^{230}$Th-$^{238}$U activity. |
| $^{230}$Th/$^{232}$Th ratio | Th230_Th232ratio | [$^{230}$Th/$^{232}$Th] ratio in the subsample in ppm. |
| $^{230}$Th/$^{232}$Th ratio error | Thratio_error | Analytical uncertainty of $^{230}$Th-$^{232}$Th ratio in ppm. |
| Uncorrected U/Th age | AgeUncorrected | Uncorrected U-Th age of the subsample in years ago. |
| Uncorrected U/Th age uncertainty | AgeUncorr_error | Analytical uncertainty of uncorrected Age in years. |
| Corrected U/Th age uncertainty | AgeCorr_error | Uncertainty of corrected age (includes Th correction) in years. |
| Initial d234U | dU234initial | Calculated initial d234U ratio in the subsample. |
| Initial d234U error | dU234i_error | Analytical uncertainty of calculated d234U initial. |
| Use in age model? | useInAgeModel | "Y" if this age constraint was used in the published age model, "N" if not. |






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

**Acknowledgments**

We gratefully acknowledge Helen Xiu, Washington University in St. Louis for the illustration in Figure 1. Iso2k is a contribution to Phase 3 of the PAGES2k Network. PAGES is supported by the US National Science Foundation and the Swiss Academy of Sciences. Support for this project includes NSF-AGS #1805141 to BLK and SS, and NSF-AGS PRF #1433408 to BLK.



**Team list**

The "Iso2k Project members" group author includes: Kerstin Braun (Institute of Human Origins, Arizona State University, Tempe, Arizona, 85287, USA), Matthieu Carré (LOCEAN Laboratory, Sorbonne Universités (UPMC)-CNRS-IRD-MNHN, Paris, 75005, France), Sylvia G. Dee (Department of Earth, Environmental, and Planetary Sciences, Rice University, Houston, Texas, 77005, USA), Alessandro Incarbona (Department of Earth and Marine Sciences, Palermo University, Palermo, 90134, Italy), Nikita Kaushal (Department of Earth Sciences, University of Oxford, Oxford, Oxfordshire, OX1

3AN, United Kingdom), Robert M. Klaebe (Department of Earth Sciences, The University of Adelaide, Adelaide, South Australia, 5005, Australia), Hannah R. Kolus (School of Earth and Sustainability, Northern Arizona University, Flagstaff, AZ, 86011, USA), P. Graham Mortyn (³³ICTA and Dept. of Geography, Universitat Autonoma de Barcelona (UAB), Bellaterra, 08193, Spain), Andrew D. Moy (³⁴Australian Antarctic Division, Kingston, Tasmania, 7050, Australia), Heidi A. Roop (Climate Impacts Group, University of Washington, Seattle, WA, 98195, USA) and Marie-Alexandrine Sicre

(LOCEAN Laboratory, Sorbonne Universités (UPMC)-CNRS-IRD-MNHN, Paris, 75005, France)

**Author contributions**

BLK directed the Iso2k Project. NPM built and managed the Iso2k database. BLK, DMT, OVC, BM, EPD, GL, SRM, EKT,

AJO, DSK, HRS, JWP, KLD, NPM, JJT designed the database (including development of metadata fields, data selection criteria). DMT, OVC, BM, EPD, GL, SRM, LJ, LCB, EKT, AJO, TO, DSK, MDJ, HRS, JWP, TJP, JJT coordinated an archive team. BLK, DMT, GMF, OVC, BM, EPD, AA, GL, SRM, LJ, LCB, EKT, AJO, TO, DSK, MDJ, HRS, MJF, HAR, JWP, MC, ADM, KLD, TJP, PGM, MAS, NPM, JJT, HRK, RMK, NK, KB assembled or entered datasets and/or metadata into database. BLK, DMT, GMF, AI, OVC, BM, EPD, AA, GL, SRM, LJ, LCB, EKT, AJO, ZK, TO, DSK, JLC, MDJ,

HRS, MJF, JWP, MC, ADM, KLD, TJP, PGM, MAS, NPM, JJT performed quality control, term standardization, database cleaning, and/or QC certification. BLK, DMT, OVC, BM, EPD, AA, GL, SRM, LJ, LCB, EKT, AJO, ZK, TO, DSK, JLC, MDJ, HRS, MJF, JWP, ADM, KLD, TJP, PGM, MAS, JJT located missing isotopic and/or chronological datasets. GMF, OVC, MDJ, MJF, NPM analyzed data and generated figures for this manuscript. BLK, DMT, GMF, OVC, BM, AA, NJA, GL, LJ, LCB, SGD, EKT, ZK, TO, DSK, SS, JLC, MDJ, HRS, MJF, MC, KLD, TJP, NPM, JJT wrote the manuscript text.

BM, NJA, and LvG coordinated with the broader 2k Network. SS and SGD helped align metadata with model comparison needs.

**Competing interests**

The authors declare no competing interests.



**Figures**

Figure 1. Schematic illustration of the global water cycle and key metadata fields in the Iso2k database. In the Iso2k database, the histories (including phase changes and transport; 'Isotope Interpretation'; red text and arrows) of different pools of environmental waters ('inferred material'; black bold text) can be inferred by interpretation of proxy records from different archives ('archive,' italic text).

1155



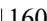

**Figure 2. The Iso2k database version 1.0. a) Spatial distribution of "primary time series" records in the Iso2k database. Symbols represent records from different archives. b) Availability of records in the Iso2k database over time during the past 2,000 years.**

Earth System Science Data Discussions Open Access

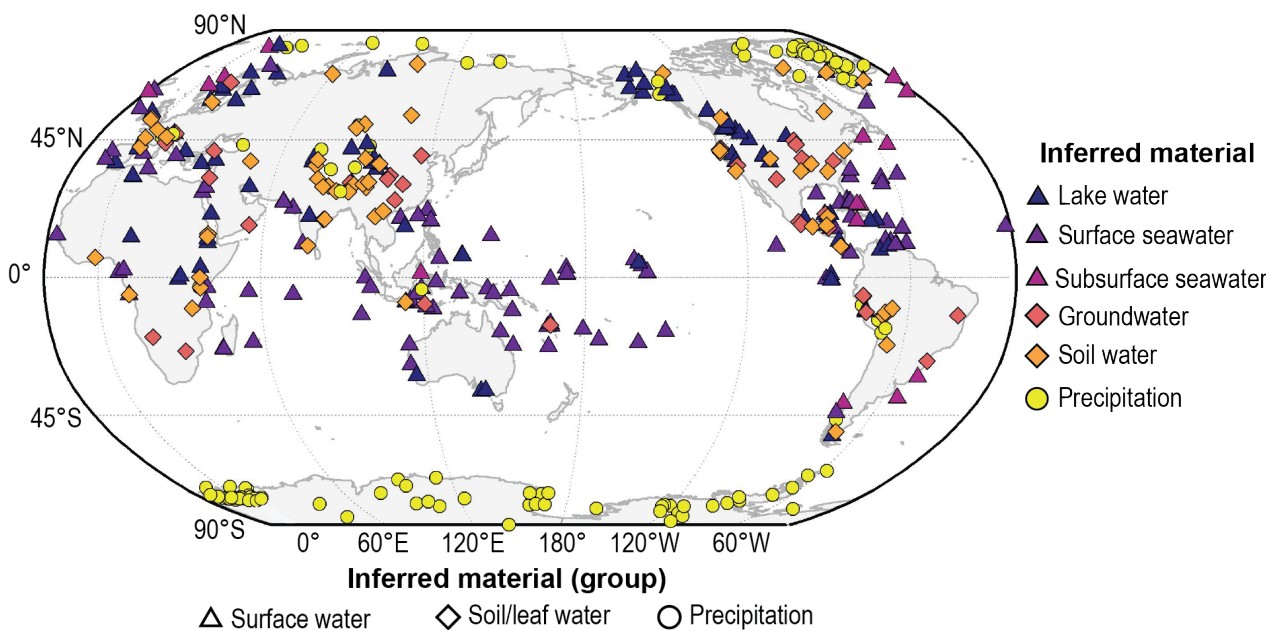

**Figure 3. Map of records in the Iso2k database with colours representing the 'Inferred Material' metadata field (Section 4.2) for each record (primary time series only; see Section 2.4). Symbols correspond to the inferred material supergroups.**

I185

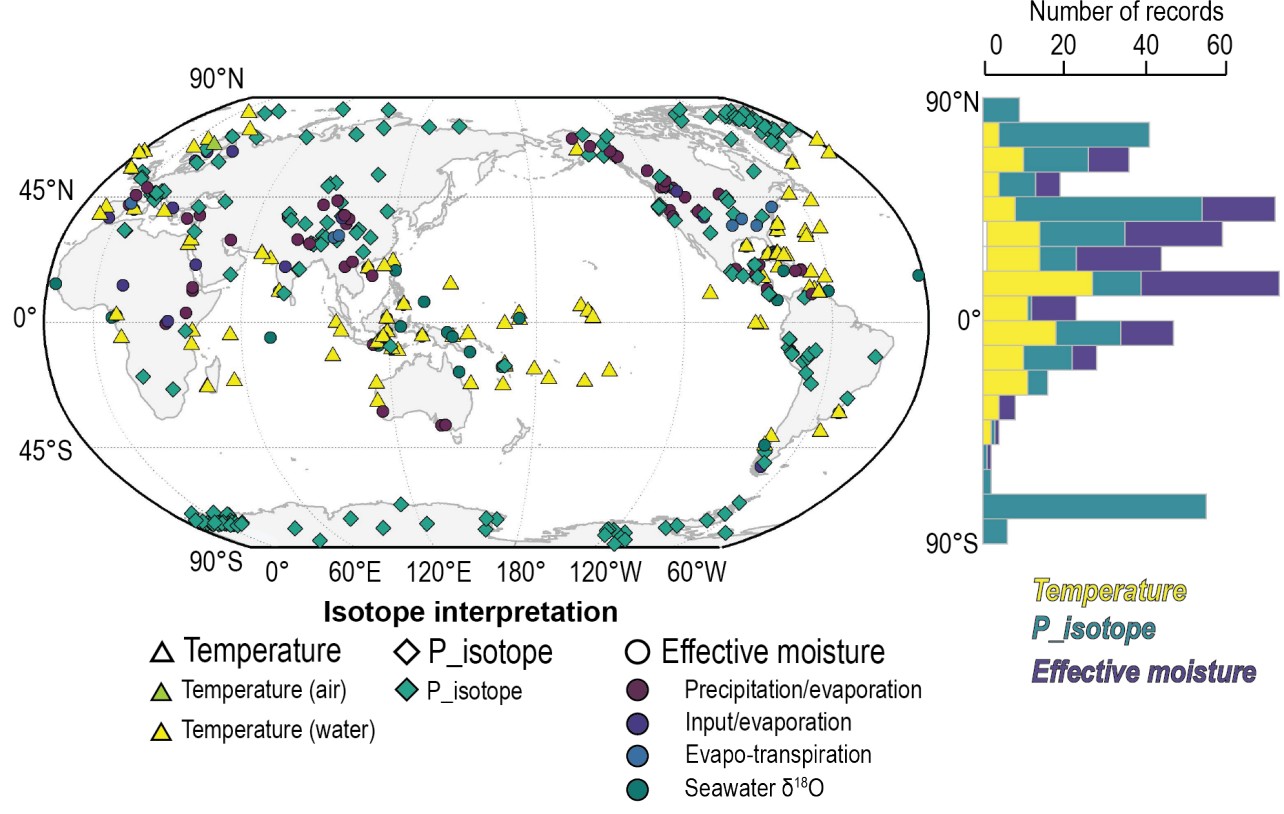

**Figure 4. a) Map of records in the Iso2k database with colours representing the first-order 'Isotope Interpretation' metadata field for each record (primary timeseries only; see Section 2.4). Symbols correspond to the three isotope interpretation 'supergroupings' (see Sections 4.3 and 5.1). b). Bar chart showing the latitudinal distribution of records in the Iso2k database. Each bar represents ten degrees of latitude.**

I190

I195

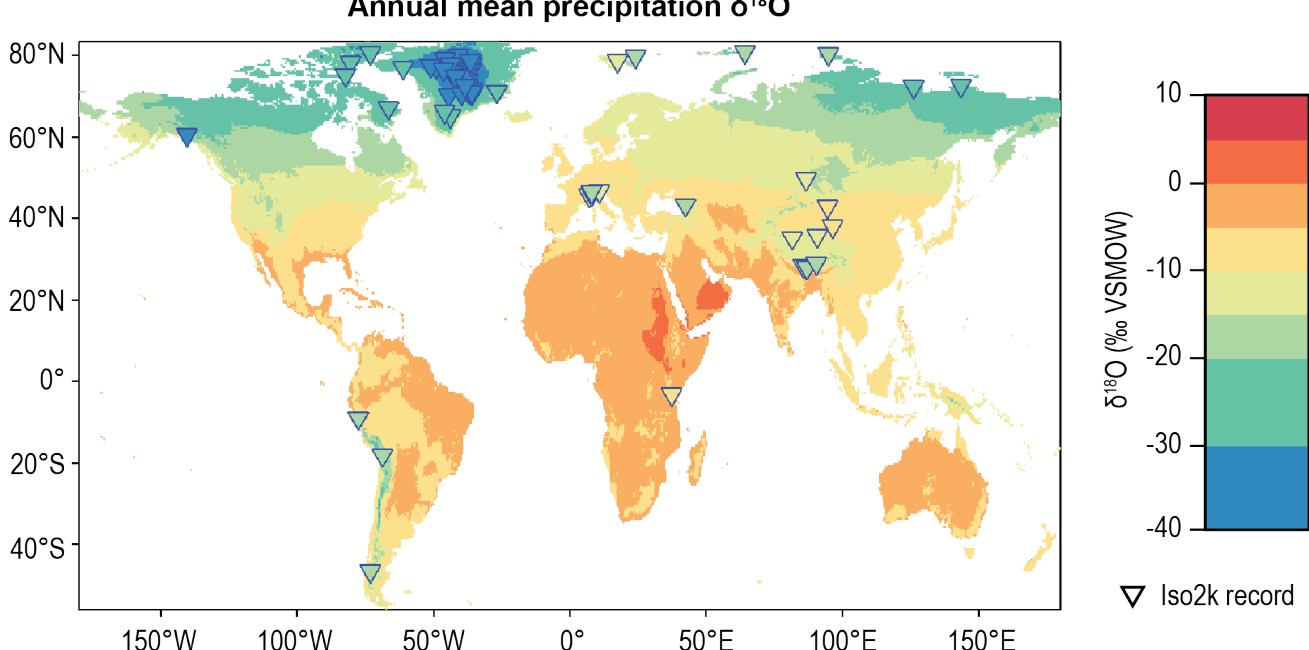

**Figure 5. Average δ¹⁸O from glacier and ground ice records in the Iso2k database (symbols), calculated as the average value since 1900 CE, compared with mean annual δ¹⁸O from the Global Network of Isotopes in Precipitation (GNIP) (shading)** (Terzer et al., 2013)**. Antarctica is excluded from this map due to the scarcity of GNIP stations.**

1200