# Peer review of "The Iso2k Database: A global compilation of paleo- $\delta^{18}\text{O}$ and $\delta^2\text{H}$ records to aid understanding of Common Era climate"

_Earth System Science Data, 2020_

## Short Comment (SC1) · 19 Feb 2020

Dear Authors,

I think you present a well prepared product worth publishing, however looking through the data I noticed a few points that could be clarified before publication.

- if I look at the 3 serialisations, I see that the R and matlab versions contain d, TS and sTS, however the python pickle only contains D and TS

```
>>> infile = open('iso2k0_14_2.pkl','rb')
```

```
>>> iso2k_dict = pickle.load(infile)
>>> iso2k_dict.keys()
dict_keys(['D', 'TS'])
```

does that have any relevance? please clarify

- directly regarding those: I have not found in the paper what D TS and sTS stand for. For the undiscerning user, this is confusing.

- your dataset at figshare makes no mention of the ESSD paper and the data are not well described at the repository. Whoever stumbles uppon your data there will be at a loss. Could you add a little data description or a readme file to the repository? The very good metadata spreadsheet you provide is also missing there.

- lastly, for final publication, it would be very helpfull to provide a short sample code for the other two serialisations as well (in a free matlab implementation like octave, and python). I don't think they would have to be as extensive as the R sample. (This may also resolve issue number one, maybe you handle the m files differently)

best regards
* * *

---

## Referee Comment (RC1) · Anonymous Referee #1 · 2 Mar 2020

The Iso2k database presented here represents a massive data synthesis effort and is a valuable contribution to our ability to effectively analyze regional and large-scale patterns in isotopic data. A random spot check of DOIs, LipD, and original data links, suggests that all the references are correct. This dataset will facilitate many new studies and will be well cited. The data quality of each individual record largely depends on the quality of the original work, but the authors are very thorough in providing the necessary metadata about each record that will allow users to evaluate the original data.

It is also good that the data links to the original author's study, since many of the

datasets in the compilation are already posted on complementary data repositories like NOAA Paleoclimatology, and includes information on the original authors' interpretation of the isotopic signal. With this as with other data synthesis products, several records may be listed in multiple data synthesis products (e.g. SISAL for speleothems).

The authors should be commended for section 6.5: we know that, for better or worse, scientific impact is measured in terms of citations, and so future work that simply cites Iso2k instead of the original studies risks undervaluing the scientific contributions of the researchers who generated the original data that forms the basis of the database, potentially influencing especially ECR career advancement. Providing the original citation information makes this easy.

The effort to compile age control points from 'dark literature' is also commendable. I was slightly concerned to see the number of especially lacustrine records where authors did not make age control data available. I wonder if there is a way to permanently host a webform for authors to submit additional information that was requested as part of this version of the database but was not provided, but could be easily included in subsequent revisions. Might be easier than direct emails. This also applies to a few records I know have come out since this paper has been posted - for these large synthesis efforts finding an efficient way to update these databases seems key given the volume of data that is published in each year.

---

## Referee Comment (RC2) · Anonymous Referee #2 · 30 Mar 2020

Konecky et al. present a large compiled dataset of isotopic tracers of the hydrologic cycle spanning the last 2000 years. This database clearly represents a huge coordinated data synthesis effort, and the authors should be lauded for their efforts to provide a standardized metadata template to facilitate intercomparison across studies and proxy types. In addition, I appreciate the authors have gone through the effort to maintain a link between datasets and the original study/citation the data were derived from. My sense is that this dataset will be highly cited, and enable new studies of Common Era hydroclimate. Furthermore, the authors have provided a roadmap for how this dataset is to be versioned and built upon; the expectation is that it will only improve in quality and utility through time. Therefore, I recommend that this study be published after a

few minor comments below are addressed.

Minor "science" comments:

1. I understand that the discussion of controls on isotope ratios of the different "archive types" in section 3 are meant to be brief, as an exhaustive discussion of controls on each proxy type would increase the length of this paper several times over! However, I would argue for a slight expansion (and correction of small errors) in the description of controls on tree-ring cellulose. First, the 27‰ offset observed by Sternberg et al. 1986 was between cellulose and water, not the cellulose precursors (L. 415-416). Cellulose synthesis from these precursors also permits exchange with water at carbonyl oxygens, so the offset between the isotope ratios of the precursors and water is likely to be different than the offset between cellulose and water, especially if the sugars are no longer in the leaf (as in tree rings). Second, I'm not sure that I understand what's meant by "as the biosynthetic fractionation is relatively constant" at L. 416 – leaf waters certainly vary in space and time rather dramatically (e.g., West et al. 2008 Plos One), and therefore, the sugars produced using these leaf waters would also have different isotope ratios. Third, some of the signal found in the leaf is dampened before being used in tree-ring cellulose as a fraction of oxygen in leaf-exported sugars exchange with xylem water in the trunk (e.g., Roden et al. 2000 GCA). Therefore, tree-ring cellulose d18O values reflect both changes in plant water sources through time (e.g., changes in xylem water isotope ratios) as well as changes in environmental conditions (e.g., more enrichment of leaf waters/sugars via a drier atmosphere, for example). Some of Paul Szejner's recent work has shown this clearly for the North American Monsoon, for example (Szejner et al. 2016 JGR).

2. L. 420 – could it also be the case that this cellulose d18O signal is due to changes in the d18O of the vapor that is the source of this precipitation? There's been a fair amount of work in the past decade that has suggested the relationship between local precipitation amount and d18O is fairly weak compared to precipitation processes (e.g., microphysics) and moisture transport history (e.g., Dayem et al. 2010 EPSL, Konecky

et al. 2019 GRL, Vimeux et al. 2011 EPSL, Bowen et al. 2019 among others)

Minor technical comments:

1. Supplemental Table S2 - rows 769-778 seem to be missing a full reference for pub1.

2. L. 207: publicly-available -> publicly available

3. L. 411. Seasonal to annual -> seasonal-to-annual

4. L. 451 - comma placement? The second half of this sentence doesn't seem to line up with the first.

5. L. 448 - seasonally-biased -> seasonally biased

6. L. 625 - what are these percentages based on? L. 621 suggests that the percentages in L. 625 should perhaps add to 100%.
* * *

---

## Editor Comment (EC1) · Attila Demény (Editor) · 2 Apr 2020

Dear Authors,

The reviews (thanks to the reviewers and to the editor) are generally positive, but there are important comments and suggestions that should be taken into consideration during the revision.

Looking forward to the revised manuscript, Attila Demény topical editor

———————————————————

---

## Author Comment (AC1) · 14 May 2020

**Editorial Comment #1:**

**The reviews (thanks to the reviewers and to the editor) are generally positive, but there are important comments and suggestions that should be taken into consideration during the revision. Looking forward to the revised manuscript**

Response to Editorial Comment #1:

We are pleased that the reviews are generally positive, and we found the feedback very helpful. In our responses to each comment, we have explained the changes that can be expected in order to address the reviewers' and editors' comments. We thank the reviewers and the editors for the constructive changes they have suggested to this manuscript.

- Bronwen Konecky
On behalf of all authors

---

## Author Comment (AC2) · 14 May 2020

**Response to SC1 (Editor Johannes Wagner):**

We thank the Editor for examining the database serializations from a user point of view and providing us with this helpful feedback. Below, we detail the revisions we will undertake in order to clarify usage of the serializations, and to provide additional sample code for non-R users. The original comment is copied below in **bold text** and our responses appear in plain text.

- Bronwen Konecky
On behalf of all authors

**I think you present a well prepared product worth publishing, however looking through the data I noticed a few points that could be clarified before publication.**

**- if I look at the 3 serialisations, I see that the R and matlab versions contain d, TS and sTS, however the python pickle only contains D and TS**
**>>> infile = open('iso2k0_14_2.pkl','rb')**

**>>> iso2k_dict = pickle.load(infile)**
**>>> iso2k_dict.keys()**
**dict_keys(['D', 'TS'])**
**does that have any relevance? please clarify**

**- directly regarding those: I have not found in the paper what D TS and sTS stand for. For the undiscerning user, this is confusing.**

We thank the Reviewer for this very helpful observation; the three variables are indeed different and require explanation. We have therefore added the following short paragraph in section 6.2 (between lines 685-690 of the Discussion paper) describing the D, TS, and sTS variables. The text is below.

The MATLAB and R serializations contain three variables: 'D', 'TS', and 'sTS.' The variable 'D' includes site-level data for each dataset structured in the LiPD format. Datasets in 'D' often contain multiple variables (e.g. stable isotope, ancillary, and chronological data), and represent how LiPD data appear when loaded into the initial environment. For most users, however, a "flattened" version of the database is more useful. We have provided this as the 'TS' variable, where each entry contains an individual time series and its associated metadata. A slightly modified version of 'TS' is included with R and Matlab, called 'sTS', which is identical to TS except that the interpretation fields are split by scope ('isotope' or 'climate') in order to simplify querying, which may be preferable for some users. The Python serialization contains only 'D' and 'TS'.

**- your dataset at figshare makes no mention of the ESSD paper and the data are not well described at the repository. Whoever stumbles uppon your data there will be at a loss. Could you add a little data description or a readme file to the repository? The**

**very good metadata spreadsheet you provide is also missing there.**

Thank you for this suggestion. Text will be added to the Figshare page with a link to the ESSD paper where users can find a description of the database, metadata fields, and supplementary tables. Links will also be added to download the 3 sample codes.

NOAA-WDS is in the process of getting a DOI minted for Iso2k version 1.0.0 (the "final publication" and official release of the database). It is our intention for that NOAA-minted DOI to be the DOI that is published in the final ESSD publication, rather than the Figshare DOI. Recognizing that v.0.14.7 of the database will be available via the Figshare DOI in perpetuity, a note will be added to that Figshare page directing users to Iso2k v1.0.0 and all future updates via the NOAA-minted DOI, and the NOAA-WDS landing page.

**- lastly, for final publication, it would be very helpfull to provide a short sample code for the other two serialisations as well (in a free matlab implementation like octave, and python). I don't think they would have to be as extensive as the R sample. (This may also resolve issue number one, maybe you handle the m files differently)**

We have created MATLAB and Python sample codes that replicate the example workflow in the original R sample code. The 3 sample codes are now available in a revised Supplementary Materials for this paper. The variable handling with Python is indeed a little different than with R and MATLAB, and the sample code now makes those procedures explicit. Thank you for this helpful suggestion!

---

## Author Comment (AC3) · 14 May 2020

**Response to Anonymous Referee #1:**

We thank the reviewer for the feedback and constructive comments on this manuscript. Below, we detail the revisions we will undertake in order to address the reviewer's comments. The original review is copied in **bold text** and our responses appear in plain text. We greatly appreciate the feedback and we believe the revised manuscript will be substantially improved.

- Bronwen Konecky
On behalf of all authors

**The Iso2k database presented here represents a massive data synthesis effort and is a valuable contribution to our ability to effectively analyze regional and large-scale patterns in isotopic data. A random spot check of DOIs, LipD, and original data links, suggests that all the references are correct. This dataset will facilitate many new studies and will be well cited. The data quality of each individual record largely depends on the quality of the original work, but the authors are very thorough in providing the necessary metadata about each record that will allow users to evaluate the original data.**

**It is also good that the data links to the original author's study, since many of the datasets in the compilation are already posted on complementary data repositories like NOAA Paleoclimatology, and includes information on the original authors' interpretation of the isotopic signal. With this as with other data synthesis products, several records may be listed in multiple data synthesis products (e.g. SISAL for speleothems). The authors should be commended for section 6.5: we know that, for better or worse, scientific impact is measured in terms of citations, and so future work that simply cites Iso2k instead of the original studies risks undervaluing the scientific contributions of the researchers who generated the original data that forms the basis of the database, potentially influencing especially ECR career advancement. Providing the original citation information makes this easy.**

We really thank the reviewer for recognizing the care we took to preserve the ability for original studies to be cited.

**The effort to compile age control points from 'dark literature' is also commendable. I was slightly concerned to see the number of especially lacustrine records where authors did not make age control data available. I wonder if there is a way to permanently host a webform for authors to submit additional information that was requested as part of this version of the database but was not provided, but could be easily included in subsequent revisions. Might be easier than direct emails. This also applies to a few records I know have come out since this paper has been posted - for these large synthesis efforts finding an efficient way to update these databases seems key given the volume of data that is published in each year.**

The Reviewer raises two important issues here, the submission of chronological data as well as the overall governance and procedure for submission of future datasets (paleo-isotopic data as well as age control data) to future versions of the Iso2k database.

We heartily agree that the submission of raw chronological data to public repositories should be standard operating procedure when authors submit datasets. Nowadays, many authors do indeed submit age control data in addition to their paleo-observation data, but for older publications this was not customary. We are very pleased to be able to now bring these chronological datasets to the publicly-available sphere.

A submission page for datasets not currently included in the Iso2k database is an excellent idea. A link to a LiPD entry template hosted through http://lipd.net/playground will be added to the official WDS-NOAA landing page for Iso2k, along with instructions for dataset submission. Because the Iso2k database was designed to use LiPD files, and the project was in direct coordination with LinkedEarth, these solutions are relatively straightforward from a technical perspective. When someone submits a new datasets as a LiPD files through http://lipd.net/playground, there is already an option to generate a "NOAA-ready" LiPD file so that users can then submit directly to the NOAA-WDS public repository, rather than having to fill out data entry templates separately. This will encourage broader submission of datasets to public repositories.

Long-term governance of the database and the procedure for community vetting and incorporation of new datasets to the Iso2k database are issues that require more than a technical fix. Those issues are currently being discussed by the group in consultation with other related PAGES groups and with the PAGES 2k coordination team. In the meantime, for purposes of this publication, we have added text to section 6.3 (line 715 of Discussion paper), with a reference to this section at the end of Section 1.3 (line 190), explaining the steps for dataset submission and stating that more detailed instructions and a link to a LiPD entry template will be added to the NOAA-WDS landing page when it is created.

**6.3 Database updates, versioning scheme, and submission of new or updated datasets**

This publication marks Version 1.0.0 of the Iso2k database (editors and reviewers: please note that you are reviewing version 0.14.7; this will become version 1.0.0 upon publication, following any edits during the review process). Following publication, the database will continue to evolve, as new datasets are added (both new studies and previous records that have been missed) and existing data or metadata are extended, or as necessary, corrected. Readers who know of missing datasets are asked to submit them directly through http://lipd.net/playground. Database users who find errors in individual datasets can submit proposed edits using the "Edit LiPD file" function at http://lipdverse.org/iso2k/current_version/, or they can use the "Report an issue" option for errors that apply to multiple datasets. More detailed instructions for dataset submission and a link to a LiPD entry template hosted through http://lipd.net/playground will be added to the WDS-NOAA landing page when they become available.

We thank the Reviewer again for these helpful suggestions.

---

## Author Comment (AC4) · 14 May 2020

**Response to Anonymous Referee #2:**

We thank the reviewer for the feedback and constructive comments on this manuscript. Below, we detail the revisions we will undertake in order to address the reviewer's comments. The original review is copied in **bold text** and our responses appear in plain text. We greatly appreciate the feedback and we believe the revised manuscript will be substantially improved.

- Bronwen Konecky
On behalf of all authors

**Konecky et al. present a large compiled dataset of isotopic tracers of the hydrologic cycle spanning the last 2000 years. This database clearly represents a huge coordinated data synthesis effort, and the authors should be lauded for their efforts to provide a standardized metadata template to facilitate intercomparison across studies and proxy types. In addition, I appreciate the authors have gone through the effort to maintain a link between datasets and the original study/citation the data were derived from. My sense is that this dataset will be highly cited, and enable new studies of Common Era hydroclimate. Furthermore, the authors have provided a roadmap for how this dataset is to be versioned and built upon; the expectation is that it will only improve in quality and utility through time. Therefore, I recommend that this study be published after a few minor comments below are addressed.**

We are pleased the reviewer found the database to be promising.

**Minor "science" comments:**

**1. I understand that the discussion of controls on isotope ratios of the different "archive types" in section 3 are meant to be brief, as an exhaustive discussion of controls on each proxy type would increase the length of this paper several times over! However, I would argue for a slight expansion (and correction of small errors) in the description of controls on tree-ring cellulose. First, the 27‰ offset observed by Sternberg et al.1986 was between cellulose and water, not the cellulose precursors (L. 415-416). Cellulose synthesis from these precursors also permits exchange with water at carbonyl oxygens, so the offset between the isotope ratios of the precursors and water is likely to be different than the offset between cellulose and water, especially if the sugars are no longer in the leaf (as in tree rings).Second, I'm not sure that I understand what's meant by "as the biosynthetic fractionation is relatively constant" at L. 416 – leaf waters certainly vary in space and time rather dramatically (e.g., West et al. 2008 Plos One), and therefore, the sugars produced using these leaf waters would also have different isotope ratios. Third, some of the signal found in the leaf is dampened before being used in tree-ring cellulose as a fraction of oxygen in leaf-exported sugars exchange with xylem water in the trunk (e.g., Roden et al. 2000 GCA). Therefore, tree-ring cellulose d18O values reflect both changes in plant water sources through time (e.g., changes in xylem water isotope ratios) as well as changes in environmental conditions (e.g., more enrichment of leaf waters/sugars via a drier atmosphere, for example). Some of Paul Szejner's recent work**

**has shown this clearly for the North American Monsoon, for example (Szejner et al. 2016 JGR).**

Thank you. We appreciate these points. The text on the biochemical fractionation has been corrected and improved. We clarify that variability in tree-ring cellulose 18O is primarily influenced by the 18O of source water and leaf water, which are influenced by climate and environmental factors. We replaced the paragraph from lines 410-421 of the Discussion paper with the new paragraph below:

The wood in tree rings (tree-ring cellulose) is one of the few terrestrial proxy archives that can be directly constrained to calendar years (McCarroll and Loader, 2004; Schweingruber, 2012). Information about climatic and environmental changes on seasonal-to-annual time scales is recorded in tree-ring cellulose $\delta^{18}$O. The $\delta^{18}$O of tree-ring cellulose is influenced by (i) the $\delta^{18}$O of source waters, and (ii) factors influencing $\delta^{18}$O of the leaf water and (iii) a fractionation related to the isotopic exchange of carbonyl oxygen of cellulose intermediates with cellular waters (derived from enriched leaf water and unaltered xylem or source waters), resulting in an overall ~27‰ enrichment of cellulose $\delta^{18}$O relative to cellular waters (Barbour et al., 2004; Gessler et al., 2014; Roden et al., 2000). This fractionation is regarded as a constant in mechanistic models (e.g., Cernusak et al., 2005; Roden et al., 2000) and so cellulose $\delta^{18}$O variability mainly reflects the $\delta^{18}$O of source water and leaf waters. The $\delta^{18}$O of the source water is closely related to the $\delta^{18}$O of precipitation-derived soil water (Bowen et al., 2019). As such, the primary climatic signal that controls $\delta^{18}$O of tree-cellulose varies by location, depending on the climatic signals controlling precipitation $\delta^{18}$O (Section 1.2). For example, tree-cellulose $\delta^{18}$O records have been interpreted to reflect temperature at mid- to high-latitude sites (e.g., Churakova (Sidorova) et al., 2019; Porter et al., 2014; Saurer et al., 2002; Sidorova et al., 2012), and precipitation amount in tropical or monsoonal sites (Brienen et al., 2013; Managave et al., 2011). As the $\delta^{18}$O of the soil water is also affected by evaporation of the soil water, precipitation minus evaporation (P-E) influences $\delta^{18}$O tree-cellulose (Sano et al., 2012; Xu et al., 2018). Water vapour pressure deficit between the leaf intercellular space and the ambient atmosphere in conjunction with the leaf physiological traits control the extent of evaporative enrichment of the source water in $^{18}$O in the leaf and hence $\delta^{18}$O of the leaf water and tree-cellulose (Kahmen et al., 2011; Szejner et al., 2016).

**Minor 'science' comment #2**

**L. 420 – could it also be the case that this cellulose d18O signal is due to changes in the d18O of the vapor that is the source of this precipitation? There's been a fair amount of work in the past decade that has suggested the relationship between local precipitation amount and d18O is fairly weak compared to precipitation processes (e.g.,microphysics) and moisture transport history (e.g., Dayem et al. 2010 EPSL, Konecky et al. 2019 GRL, Vimeux et al. 2011 EPSL, Bowen et al. 2019 among others).**

We agree that cellulose d18O can reflect factors other than temperature and precipitation amount, such as moisture source, precipitation microphysics, and transport history. We revised the text here to clarify that the climatic controls on the d18O of cellulose are largely driven by the climatic controls on d18O of precipitation, which are discussed in Section 1.2. See the above revised paragraph. We also added a reference to section 1.2 and also added the reviewer's suggested citations to that section.

**Minor technical comments:**

**1. Supplemental Table S2 - rows 769-778 seem to be missing a full reference for pub1.**

These rows have a full reference for pub1 (Moreno et al., 2012, QSR, complete with doi). However, the formatting of special characters appears to have been incorrect, which may have led to problems viewing the publication reference. We fixed this as well as other special character errors in Supplemental Table S2.

**2. L. 207: publicly-available -> publicly available**

Fixed.

**3. L. 411. Seasonal to annual -> seasonal-to-annual**

This line was edited for clarity.

**4. L. 451 - comma placement? The second half of this sentence doesn't seem to line up with the first.**

The second half of the sentence L. 451 was rephrased. "...foraminifera calcite is systematic **i.e.** the δ18Osw can be reconstructed…"

**5. L. 448 - seasonally-biased -> seasonally biased**
Changed

**6. L. 625 - what are these percentages based on? L. 621 suggests that the percentages in L. 625 should perhaps add to 100%**

This is indeed a bit confusing as written. Therefore, we clarified the text in this paragraph so that the percentages are out of stable isotope records, rather than being out of all the records in the database (which included 255 ancillary records).

We thank the Reviewer again for these helpful suggestions.

---

## Author Response (AR2)

7 July 2020

Dear Dr. Demény,

We are pleased that our manuscript has been accepted for publication in ESSD pending very minor corrections. Attached please find the corrected manuscript. We made all suggested changes. Please let us know if there are any further questions or edits needed.

Thank you very much for your handling of this paper!

Best,
Bronwen Konecky
Washington University, USA

(On behalf of all co-authors)

[revised manuscript text omitted]

**Legend:** Key metadata fields in the Iso2k Database

*archive*

**inferred material**

Water cycle processes affecting $\delta^{18}O$, $\delta^2H$ (Isotope Interpretation):

- - ▸ phase change (evaporation, condensation)
— ▸ transport, infiltration, runoff

**Figure 1. Schematic illustration of the global water cycle and key metadata fields in the Iso2k database. In the Iso2k database, the histories (including phase changes and transport; 'Isotope Interpretation'; red text and arrows) of different pools of environmental waters ('inferred material'; black bold text) can be inferred by interpretation of proxy records from different archives ('archive,' italic text). Base illustration by Helen Xiu, Washington University.**

[Figure]

**Figure 2. The Iso2k database version 1.0.0. a)** Spatial distribution of "primary time series" records in the Iso2k database. Symbols represent records from different archives. **b)** Availability of records in the Iso2k database over time during the past 2,000 years.

[Figure]

**Figure 3. Map of records in the Iso2k database with colours representing the 'Inferred Material' metadata field (Section 4.2) for each record (primary time series only; see Section 2.4). Symbols correspond to the inferred material supergroups.**

[Figure]

**Figure 4. Left: Map of records in the Iso2k database with colors representing the first-order 'Isotope Interpretation' metadata field for each record (primary timeseries only; see Section 2.4). Symbols correspond to the three isotope interpretation 'supergroupings' (see Sections 4.3 and 5.1). Right: Bar chart showing the latitudinal distribution of records in the Iso2k database. Each bar represents ten degrees of latitude.**

[Figure]

**Annual mean precipitation δ$^{18}$O**

90      **Figure 5.** Average δ$^{18}$O from glacier and ground ice records in the Iso2k database (symbols), calculated as the average value since 1900 CE, compared with mean annual δ$^{18}$O from the Global Network of Isotopes in Precipitation (GNIP) (shading) (Terzer et al., 2013). Antarctica is excluded from this map due to the scarcity of GNIP stations.

95